# A 3D ray traced biological neural network learning model

Brosnan Yuen[1], Xiaodai Dong [1] ✉ & Tao Lu [1] ✉

Training large neural networks on big datasets requires significant computational resources and time. Transfer learning reduces training time by pretraining a base model on one dataset and transferring the knowledge to a new model for another dataset. However, current choices of transfer learning algorithms are limited because the transferred models always have to adhere to the dimensions of the base model and can not easily modify the neural architecture to solve other datasets. On the other hand, biological neural networks (BNNs) are adept at rearranging themselves to tackle completely different problems using transfer learning. Taking advantage of BNNs, we design a dynamic neural network that is transferable to any other network architecture and can accommodate many datasets. Our approach uses raytracing to connect neurons in a three-dimensional space, allowing the network to grow into any shape or size. In the Alcala dataset, our transfer learning algorithm trains the fastest across changing environments and input sizes. In addition, we show that our algorithm also outperformance the state of the art in EEG dataset. In the future, this network may be considered for implementation on real biological neural networks to decrease power consumption.

In artificial neural networks, many models are trained for a narrow task using a specific dataset. They face difficulties in solving problems that include dynamic input/output data types and changing objective functions. Whenever the input/output tensor dimension or the data type is modified, the machine learning models need to be rebuilt and subsequently retrained from scratch. Furthermore, many machine learning algorithms that are trained for a specific objective, such as classification, may perform poorly at other tasks, such as reinforcement learning or quantification.

Even if the input/output dimensions and the objective functions remain constant, the algorithms do not generalize well across different datasets. For example, a neural network trained on classifying cats and dogs does not perform well on classifying humans and horses despite both of the datasets having the exact same image input[1]. Moreover, neural networks are highly susceptible to adversarial attacks[2]. A small deviation from the training dataset, such as changing one pixel, could cause the neural network to have significantly worse performance. This

problem is known as the generalization problem[3], and the field of transfer learning can help to solve it.

Transfer learning[4–10] solves the problems presented above by allowing knowledge transfer from one neural network to another. A common way to use supervised transfer learning is obtaining a large pre-trained neural network and retraining it for a different but closely related problem. This significantly reduces training time and allows the model to be trained on a less powerful computer. Many researchers used pre-trained neural networks such as ResNet-50[11] and retrained them to classify malicious software[12–15]. Another application of transfer learning is tackling the generalization problem, where the testing dataset is completely different from the training dataset. For example, every human has unique electroencephalography (EEG) signals due to them having distinctive brain structures. Transfer learning solves the generalization problem by pretraining on a general population EEG dataset and retraining the model for a specific patient[16–20]. As a result, the neural network is dynamically tailored for a specific person and can

[1]Department of Electrical and Computer Engineering, University of Victoria, 3800 Finnerty Road, Victoria V8P 5C2 BC, Canada. ✉e-mail: xdong@ece.uvic.ca; taolu@uvic.ca

interpret their specific EEG signals properly. Labeling large datasets by hand is tedious and time-consuming. In semi-supervised transfer learning[21–24], either the source dataset or the target dataset is unlabeled. That way, the neural networks can self-learn which pieces of information to extract and process without many labels.

For comparing the advantages and disadvantages of the related works, we have created Table 1 in Supplementary Material Section S.2[25] to showcase the features of each research article. Among them are transfer learning with neural AutoML[26], two-stage evolutionary neural architecture search[27], and a self-adaptive mutation neural architecture search algorithm based on blocks[28]. Most of the neural evolution algorithms in the literature use discrete blocks or layers to construct networks. Architectures using discrete blocks are highly restrictive because only a select few layers are compatible with the existing layers. If the optimal architecture uses blocks that are incompatible with the current blocks, then the current network can not be transferred into the optimal architecture. Moreover, when the input/output dimension changes, the input/output layer is deleted and replaced with a new layer that matches the new dimensions. Deleting old layers impedes transfer learning because the old weights are not transferred to the new network. This increases training time as the new layers are trained from scratch.

On the other hand, bio-inspired artificial neural networks take advantage of neuron positions to generate new neural connections and offer far more flexibility in solving unseen problems/datasets. In place of having separable discrete layers and organized connections,

NeuCube[29] arranges neurons in a cube lattice and randomly creates neural connections based on relative neuron distances. Neurons close together have a higher probability of forming new connections, while neurons further apart have a lower probability. Moreover, the algorithm also generates long-distance connections, which reduces the degree of separation between any two neurons and improves performance. Going further, HyperNEAT[30] and DES-HyperNEAT[31] use both absolute and relative neuron positions to determine neural connectivity and the overall architecture. For every combination of two neurons, their three-dimensional (3D) positions are fed into a CPPN that predicts the values of the weights. The flexible connectivity enables HyperNEAT to handle changing input and output dimensions, while also growing and shrinking hidden neurons at will. However, NeuCube and HyperNEAT do not support the ability to join and merge multiple neural networks together. This prohibits the ability to scale to very large neural networks by joining multiple smaller neural networks together. Furthermore, those implementations do not support sparse matrices, which deliver the same performance but with less training time and memory usage in very large networks. Moreover, their activation functions are fixed and are not flexible enough to suit different datasets. Incorporating neuroplasticity mechanisms from real biological neural networks could solve these problems.

Real biological neural networks consist of two primary classes of cells: glial cells and neurons. Neurons are made out of axons, axon branches, synapses, dendrites, and soma. As an example, Fig. 1a shows a typical central nervous system[32]. The red, blue, and green colors

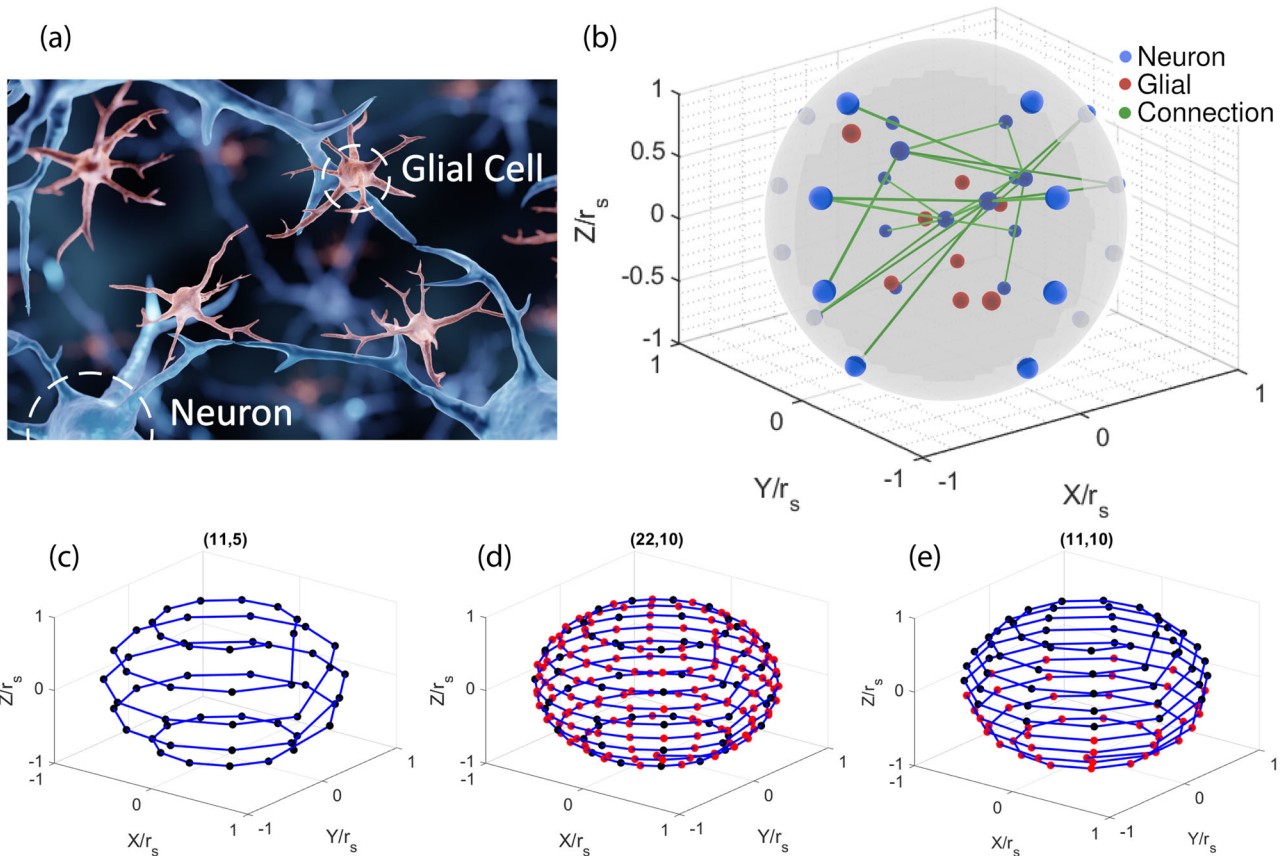

Fig. 1 | Overview of the ray-traced biological neural network. a An illustration of a central nervous system[32]. b Our simulated three-dimensional (3D) ray-traced biological neural network (RayBNN). The blue, red, and green colors correspond to neurons, glial cells, and axons, respectively, where the axons carry electrical and neurotransmitter signals from one neuron to another. c Location of input neurons at the surface of the network sphere. A two-dimensional (2D) ordered data such as images can be mapped to the neurons with order preserved. The blue line connects the neurons into a one-dimensional (1D) array if the data is 1D. d Transfer of input neurons to a new network sphere where the dimension of the data is densified. The red dots are the new input neurons, and the location of the old neurons (black dots) are not changed. e If the new dataset concatenates the old dataset, then the old neurons migrate to the north while new neurons are created in the south of the new network sphere. Note that all neurons occupy the same solid angle and access the hidden neurons underneath without bias.

correspond to glial cells, neurons, and axons, respectively, where the axons carry electrical and neurotransmitter signals from one neuron to another. Each neuron only has one axon, but it can split into multiple axon branches, which allows the neurons to output neurotransmitters and electrical signals to multiple neurons. In order to form a new neural connection, the axon branches move towards neurotransmitters emitted by other neurons until it connects to a dendrite. Afterward, the axon can send signals through the synapses on the dendrites to reach the somas of the other neurons. However, glial cells or dead neurons could block the paths of axons, preventing them from attaching to other dendrites. The cell body of the neuron is called the soma, and it collects the net charge of the neurotransmitters and electrical signals from the dendrites. If the soma's voltage exceeds a threshold, the soma fires a pulse exiting from the axon. The main purpose of the glial cells is to insulate neurons from each other and the extracellular fluid. This prevents signals from leaking into the extracellular fluid or firing unintended neurons. A more detailed explanation is available in the Supplementary Material Subsection S.1[25].

## Results

### Proposed algorithm for simulating biological neural networks

Mimicking the BNN, we propose 3D ray-traced biological neural networks (RayBNN), as shown in Fig. 1b to solve limitations in transfer learning. Our RayBNN is constructed by uniformly distributing hidden neurons and glial cells within a 3D neural network sphere, where the cells do not intersect. Upon setting up the positions of the hidden neurons and glial cells, we then assign the positions of the input and output neurons. Some datasets have images as inputs. In those cases, the input neurons are evenly placed onto the sphere surface in order to preserve the relative distances between pixels. On the other hand, the output neurons are all fixed at the origin, similar to the architecture of a human brain. Naturally, this allows output neurons to pool and aggregate information from hidden neurons as the neural connections condense at the center of the sphere. To retain the order of the input data, we assign input neurons to the sphere surface according to the "Cell location assignment and distribution analysis" section. As shown in Fig. 1c, the input neurons are arranged at the surface so that the order of one-dimensional (1D), two-dimensional (2D), and 3D data features will be retained through direct mapping. Further, each neuron occupies the same solid angle at the sphere surface so that all input neurons can connect to hidden neurons underneath without bias. Moreover, the sphere architecture enables output synchronization, as the distance between any input neuron and output neuron is the same. When the model is transferred to a new dataset with densified input dimensions, new neurons (red dots) can be inserted in between old neurons, as shown in Fig. 1d, without the need to move the old neurons. This is suitable for, e.g., transferring learning to higher-resolution images. On the other hand, if the increased data feature is to be concatenated to the previous data features, then as shown in Fig. 1e, our algorithm can migrate old neurons toward the north of the sphere while the new neurons are added to the south without changing the data feature order.

We further create unidirectional connections between neurons that have line-of-sight using raytracing algorithms discussed in the "Forming neural connections via raytracing" section. Glial cells, just like in real BNNs, are functioned as objects to block connections between neurons that are too far apart, which reduces overfitting in our learning model. We store the weight of every unidirectional neural connection inside a sparse matrix, which enables the RayBNN to transform into any architecture without needing to resize the matrix. Additionally, a universal activation function (UAF) outlined in the "Universal activation function" section is deployed to every neuron to enable the activation functions to evolve during the knowledge transfers.

Using the advantages of our RayBNN, we can adapt and transfer the network to any arbitrary architecture. For example, large neural networks take a long time to train. To solve the problem, we start with a small neural network that trains very fast and then transfer the knowledge to a much larger network, reducing training time. During the transfer, the number of neurons increases. As a result, we add more neurons to the 3D network sphere, and ray trace new neural connections while preserving the old connections. The network sphere size may increase accordingly to keep the neuron collision rate unchanged. Moreover, the UAF adapts its activation functions to suit more neural connections and neurons. On the other hand, if the new dataset requires fewer neurons and connections. The RayBNN can delete neural connections biased towards those having the smallest absolute valued weights because they have the least impact. Some of the neural connections can be redistributed across other neurons. Afterward, unused neurons are pruned to improve efficiency and accuracy.

RayBNN is very similar to real-life biological neural networks due to having 3D physical cell locations, line-of-sight neural connectivity, signal propagation delays, glial cells, cell growth, cell death, neural network merges, and neural network bifurcations. Firstly, both the RayBNN and the real-life BNN are physically constrained by the radius of the entire neural network, cell radii, and cell density. For a neural network radius, there is a finite amount of cells within the volume because the cells can not be closer than 2 cell radii. Due to those physical constraints, both RayBNN and real-life BNN have line-of-sight neural connections that can be blocked by glial cells or other neurons. Subsequently, RayBNN has a signal propagation delay that is similar to a real-life BNN because it takes time for information to travel from one neuron to another. Real-life BNN has glial cells to inhibit or electrically isolate neurons from each other to prevent infinite signal loops or neuron overfiring. With the same idea, we implemented glial cells in our RayBNN to reduce neural connections and prevent overfitting of the network. Similar to real-life BNN, our RayBNN can dynamically grow or shrink by adding new neurons or deleting neurons. Moreover, our RayBNN can join or merge multiple neural networks along multiple axes. This has a higher degree of connectivity between blocks than traditional artificial neural networks and results in better integrations.

### Hyperparameter tuning and model characterization

Our model does not allow two cells (neuron or glial cell) to intersect each other, and deleting them is costly. Therefore, we characterize our model in Fig. 2 by first determining the cell density ($\eta$) to keep the probability of a cell collision ($P_c$) low. Afterward, given the predetermined number of cells based on the dataset complexity, we calculate the network sphere radius ($r_s$) using the selected cell density and locate cells within the sphere radius, where we verify the uniform distribution of cells. Subsequently, we ray trace neural connections and plot the probability density functions of the connection lengths $P_{nc}(r)$ and the number of connections per neuron ($N_c$).

For our model, we set $P_c < 1\%$. To achieve this, we first adopt 240,000 neurons and an equal number of glial cells and vary the sphere radius to plot the collision probability vs. cell density as blue dots with error bars in Fig. 2a. The log least-square fitting of the data (blue dashed line) results in a slope of 1.06, indicating the almost linear dependency between the probability and the density, which is also confirmed analytically (Eq. (7), red solid line in the plot) in "Cell collision detection and analysis" Section. As shown, to reduce $P_c$ below the 1% threshold, the cell density $\eta$ that takes into account both neuron and glial cells can not exceed $2.83 \times 10^{-4} r_n^{-3}$, leading to a minimum network sphere radius of $r_{s,\min} = 739.81 r_n$, where $r_n$ is the radius of a neuron/glial cell. These values are in close agreement with the theoretical predicted maximum density $\eta < \frac{3 P_c}{32 \pi r_n^3} = 2.98 \times 10^{-4} r_n^{-3}$ and minimum sphere radius $r_s > 726.8 r_n$, according to Eq. (7).

In Fig. 2b, we further compare the computation time of three collision detection algorithms. Shown as red dots with error bars, the computation time for the serial algorithm, of which one cell is checked at a time, grows linearly with the number of cells in the sphere

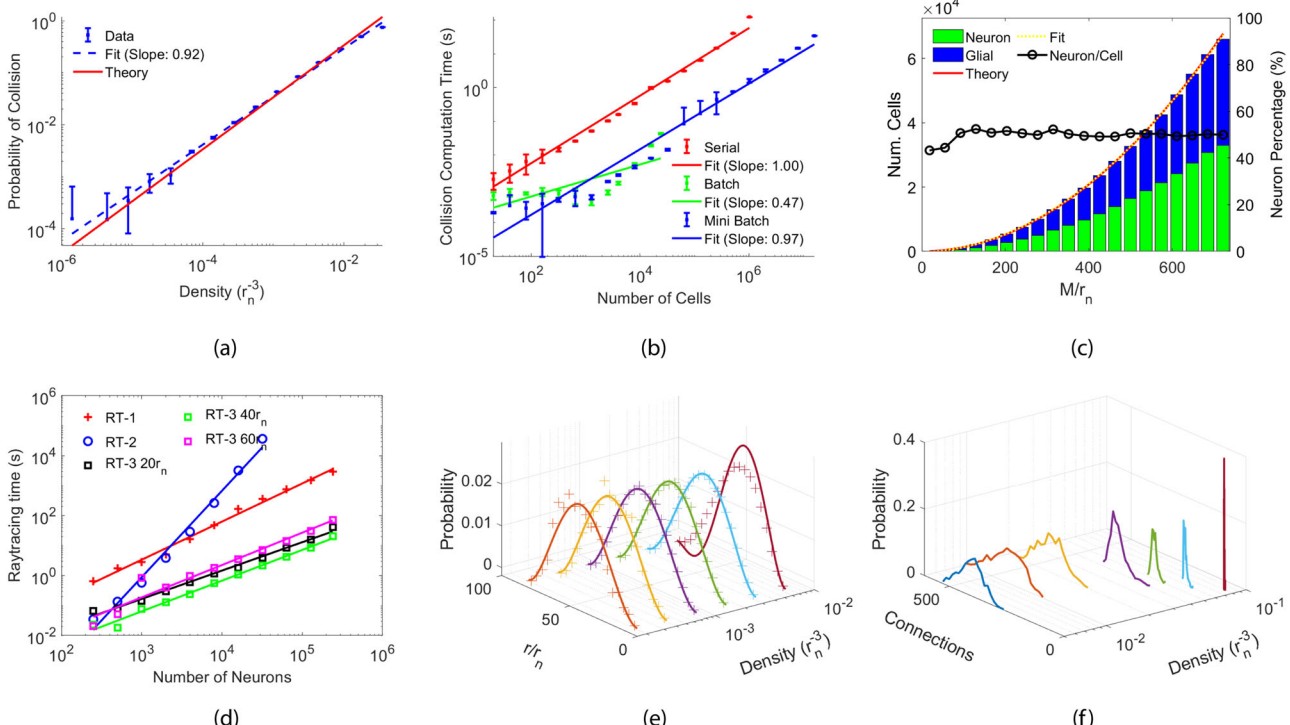

**Fig. 2 | Characterization of RayBNN. a** The probability of a cell collision versus the density of the sphere. **b** The collision detection time versus the total number of cells in the sphere. **c** The number of cells as a function of distance from the sphere center. The red bars represent neurons, while the blue bars represent glial cells. **d** Raytracing time as a function of the number of neurons in the network. The red, blue, black, green, and magenta lines represent RT-1, RT-2, RT-3 $20r_n$, RT-3 $40r_n$, and RT-3 $60r_n$, respectively. **e** The probability distribution of the neural connection length at various densities is shown as plus markers. The solid lines of the same color are the theoretic results. **f** The probability distribution of the number of neural connections per neuron.

according to the least-square fitting (red solid line with a slope of 1.00). It takes 40 s for all 480,000 cells, which is slow. The batch algorithm, shown as the green dots with error bars, in which every cell is checked at the same time, is much faster. The least-square fitting (green solid line with a slope of 0.47) confirms that the computation time only grows at a rate proportional to the square root of cell number ($N$). However, it requires $O(N^2)$ memory to set up a $N \times N$ matrix, which crashes for large amounts of cells. To solve this, we implement a mini-batch algorithm (blue dots with error bars) that takes less memory and checks 480,000 cells in 0.68 s, although it has the same growth rate (blue solid line with a slope of 0.97) as the serial method.

Using the calculated $r_s$, we assign all cells uniformly to the neural network sphere according to the procedure described in the "Cell location assignment and distribution analysis" Section. The histograms in Fig. 2c show the number of neurons (green bars) and glial cells (blue bars) as functions of distance from the network sphere center. The perfect parabolic fitting to cell histograms (yellow dashed line) shows the number of cells quadratically increases with distance. The quadratic dependency is in perfect agreement with the theoretic prediction of Eq. (4), which is shown as the red solid line in the plot and confirms the cells are uniformly distributed across the network sphere. Moreover, neuron percentage is almost constant at 50% other than expected fluctuations in low count bins because there are equal numbers of neurons and glial cells. Therefore, it is confirmed that our algorithm did distribute cells uniformly within the sphere.

After generating the positions of cells, we employ raytracing algorithms to create neural connections between neurons. In the "Forming neural connections via raytracing" Section, we present three raytracing algorithms for creating connections. As shown in Fig. 2d, RT-1 (red plus markers) does not scale well because it requires a large number of rays per neuron in order to establish connections between neurons unblocked by glial cells. Using

10,000 rays per neuron in our current 480,000-cell network, it takes 2891 seconds to generate connections for neurons they hit. The least-square fitting of the log of raytracing time and the number of neurons (red line) shows a slope of 1.27, suggesting its computational complexity of $O(N^{1.27})$. On the other hand, as expected, RT-2 (blue circles) is also slow as it requires 36,663 s for 32,000 neurons and needs $O(N^3)$ comparisons according to the least-square fitting (blue solid line, with a slope of 2.88). To reduce the number of comparisons, we adopt RT-3 (black squares) that only connects all neurons within a fixed sphere radius $r_{RT}$. The black, green, and pink squares in the plot are the results for a radius of $20r_n$, $40r_n$, and $60r_n$, respectively. As shown from the least-square fittings (solid lines with the same color), although RT-3 has the same complexity as RT-1, it runs much faster due to the reduced number of comparisons per neuron. In particular, at $r_{RT} = 40r_n$, these distance-limited rays significantly reduce raytracing time to 20 s for all 240,000 neurons.

The plus markers in Fig. 2e show the probability of forming a neural connection compared to the neural connection length normalized to $r_n$ using RT-3 as a raytracing method. In this figure, the RT-3 radius is $40r_n$ and has a diameter of $80r_n$. Consider a single neuron that is the starting point for a ray. When the ray moves further away from the starting neuron, the number of cells for the ray to terminate increases exponentially. This is reflected in Fig. 2e, where the probability of forming a neural connection increases quadratically as the neural connection length increases linearly. The quadratic relationship holds until the neural connection length reaches close to the network radius, where the probability of forming a neural connection peaks. Afterward, the probability decays because the neurons outside of the $40r_n$ radius sphere are prohibited to connect. Moreover, the probability of forming a neural connection is zero when the neural connection length is greater than the diameter of the neural network

sphere. At sufficiently low density, the neuron length distribution is nearly unchanged by the density. The probability distribution is further confirmed from theoretic analysis under low-density approximation detailed in the "Neural connection length probability distribution function" section and displayed as solid lines with the same color. As shown, at a low density of $1.6 \times 10^{-4} r_n^{-3}$ (red color), the simulation data (plus markers) displays large fluctuation due to low cell counts inside the cluster sphere, and at the largest density of $0.0205 r_n^{-3}$ (purple line), the theoretical model does not match the probability well as the low-density approximation is no longer satisfied. Meanwhile, at the densities in between, the theoretic model is in close agreement with the simulation.

Figure 2f shows that the number of neural connections also changes with the density. When the density is low at $5 \times 10^{-3} r_n^{-3}$ (blue solid line), the number of neural connections per neuron is 400. As the density increases to $0.01 r_n^{-3}$ (red solid line), $0.02 r_n^{-3}$ (yellow solid line), and $0.04 r_n^{-3}$ (purple solid line), the number of connections per neuron drops to 300, 200, 150, and finally at $0.08 r_n^{-3}$ (dark red line) to around 15 connections per neuron. This is due to glial cells and the other neurons blocking the number of connections when the neural network is very dense.

## Alcala dataset

The proposed biological neural networks are useful for many different types of transfer learning applications and datasets. Objectively, we aim to reduce training time by transferring weights from a smaller neural network to a larger network. For a simple 1D example, we used the Alcala Tutorial 2017 dataset[33–35] for wireless indoor localization. The objective is to predict the positions of wireless devices given the received signal strength intensity (RSSI) of the Wi-Fi access points (APs). Each AP provides one input RSSI feature, where a value of −99 dBm indicates the AP is far away, while a value of −1 dBm indicates the AP is nearby. Furthermore, an RSSI value of +100 dBm implies the AP is not detected at all. The neural networks have to use the RSSI values and the APs' positions to predict the X and Y positions of the wireless devices.

To simulate this, we started with six APs as our initial training dataset and built our initial RayBNN upon it. The initial RayBNN has six input neurons and two output neurons. Although the number of hidden neurons can be determined through a standard hyperparameter tuning process, we here empirically set it to 40. Correspondingly, we assign an equal number of glial cells to mimic the real biological neural network, although it can also be tuned if necessary. With the prescribed algorithm in the "Cell collision detection and analysis" section, the network sphere is set to $r_s = 42 r_n$ to keep the collision rate below 1%. Consequently, through the RT-3, 1800 connections are created with a total of 5300 trainable parameters. After training, we increased the dimension of the new training dataset empirically to eight APs and transferred the trained model to the new dataset. As every AP provides one input feature, the number of neural network inputs of the new dataset increases along with the model complexity. Therefore, we increased the network to eight input neurons. Following the same procedure as the previous iteration, we also increase the network to 50 hidden neurons and 50 glial cells, while adjusting the network sphere to $r_s = 45 r_n$ accordingly. Meanwhile, 5700 new connections are also created before training, leading to the total number of parameters to 11,000. As shown in the red circles with a solid red line in Fig. 3a, this process continued until the network reached the maximum input feature size of 162.

After training the RayBNN for the Alcala dataset, we plot the network characteristics in Fig. 4. The RT-3 radius controls the maximum neural connection length and indirectly limits the neural connectivity/number of connections. To find the lowest MAE and fastest training time, we sweep the RT-3 radius in Fig. 4a. As shown in the figure, the MAE reaches a minimum of $60 r_n$ with a training time of 70 s. Figure 4b displays the probability density function of the weighted adjacency matrix. The least-square fit to Gaussian indicates the PDF roughly follows zero-mean Gaussian with the standard deviation normalized to the maximum weight value in magnitude $\sigma = 0.039$, where the majority of the weights are centered around the normalized mean of $\mu = -0.002$. According to the distribution of deleted values in Fig. 4c, probabilistically deleting 5% of the smallest weights removes many zero-valued weights at a high probability, while also deleting large valued weights at a low probability. Overall, as shown in Fig. 4d, the weighted adjacency matrix is quite sparse, with the sparsity dropping to below 40% at 162 APs. Therefore, our implementation of a sparse matrix enhances memory usage efficiency substantially.

A snapshot of 300 neuron activation functions is pictured in Fig. 4e, while the animation of UAF evolution can be found in Supplementary Movie 1, "RayBNN evolution". Similar to the weights, the old activation functions are reused and adapted to the new problem every time transfer learning is invoked. This reduces training time as the old activation functions are pre-trained for the new problem. Figure 4f displays the evolution of the weighted adjacency matrix across multiple knowledge transfers. Unlike the other transfer learning methods, the biological neural network does not delete the input layer or the output layer. Instead, it expands the weighted adjacency matrix with new weights while keeping the old weights every time the neural network is transferred to a new dataset or the input/output dimension changes.

In order to compare the performance to the biological neural networks, we trained CNN, GCN2, LSTM, MLP, GCN2LSTM, and BiLSTM models with the same method as above. Details of the model configurations can be found in the "Details of other models for comparison" section. As shown in Fig. 3a, the trainable parameters of our RayBNN (red circle with solid line) increase at a much lower rate compared to other methods, possibly due to our efficient deletion of redundant neurons and connections to keep the network compact. Individual segment training times are shown in Fig. 3b. Consequently, at the final learning stage with all 162 APs included, our RayBNN demonstrated an 11.4 s segment training time and 73.2 s cumulative training time off (red solid lines with error bars in Fig. 3b and c). In contrast, the second fastest algorithm, BILSTM, reaches 48.0 s in segment training and 506s in cumulative training time (purple lines in Fig. 3b, c), which are more than 4× and 7× slower than RayBNN. The proposed RayBNN is far faster in transferring knowledge from one problem to another similar problem.

The RayBNN does not only run faster, but it also is more accurate in determining location. The neural network performances on the Alcala Tutorial 2017 dataset are shown in Fig. 3d. When the number of APs/inputs increases, the mean absolute value (MAE) decreases due to the neural networks having more information about the wireless device's location. Among all models, RayBNN reaches the lowest MAE of 0.89 m at 162 APs, while the MAE of the rest models varies between 0.95 to 1.33 m. For the specific 162 AP result, we plot the probability distribution function (PDF) in Fig. 3e and the cumulative distribution function (CDF) in Fig. 3f. For RayBNN, the most probable error is 1.1 m, and at 80% CDF errors are below 2 m, both are among the lowest in all models.

## EEG motor-imagery dataset

In EEG datasets, the objective is to retrieve information from the subject's brain using multiple electrodes placed on the subject's head/brain. However, every human has a unique set of EEG signals that is completely different from every other person. This is due to having distinct brain structures and electrode placements. As a consequence, most algorithms are unable to perfectly generalize across different subjects, especially if they have not seen the subject's specific waveforms before.

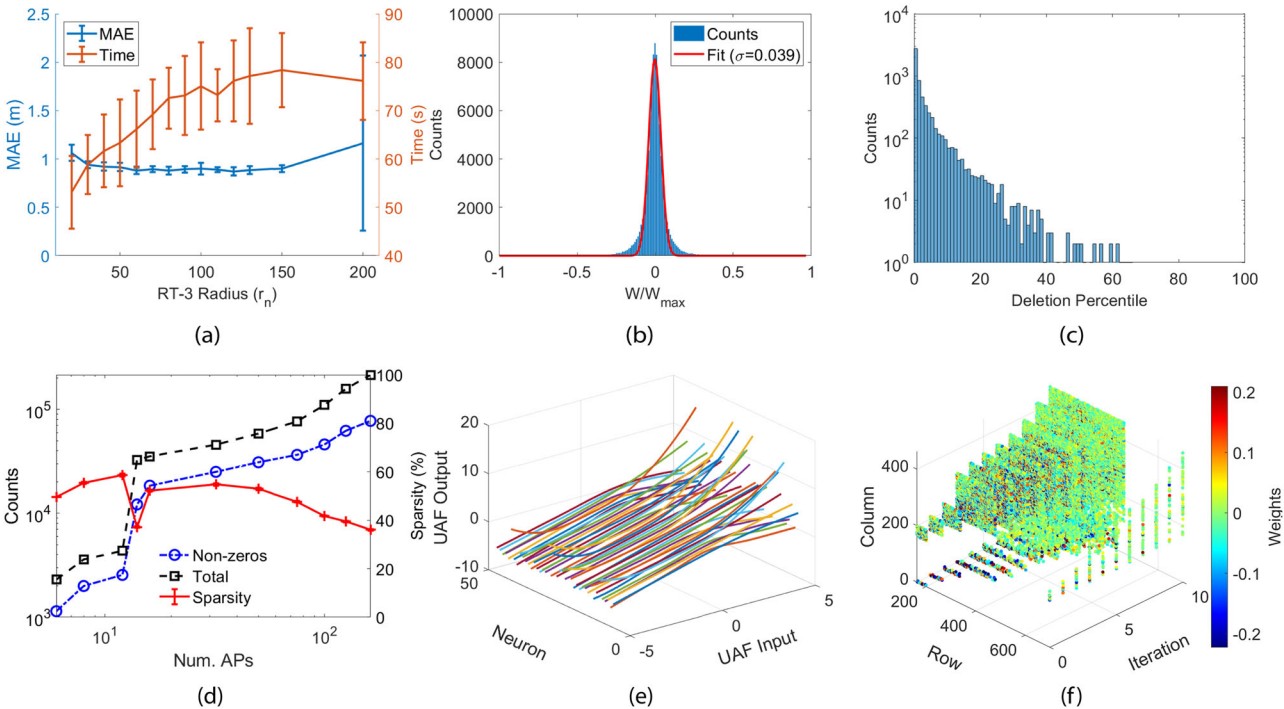

**Fig. 3 | Comparisons of various algorithms on the Alcala Tutorial 2017 dataset**[33] **with tenfold testing. a** Trainable parameters vs **b** segment training time. **c** Cumulative training time across a number of APs/inputs. **d** MAE of the various algorithms across different numbers of APs. **e** Probability distribution function of the localization error. **f** Cumulative distribution function of the localization error.

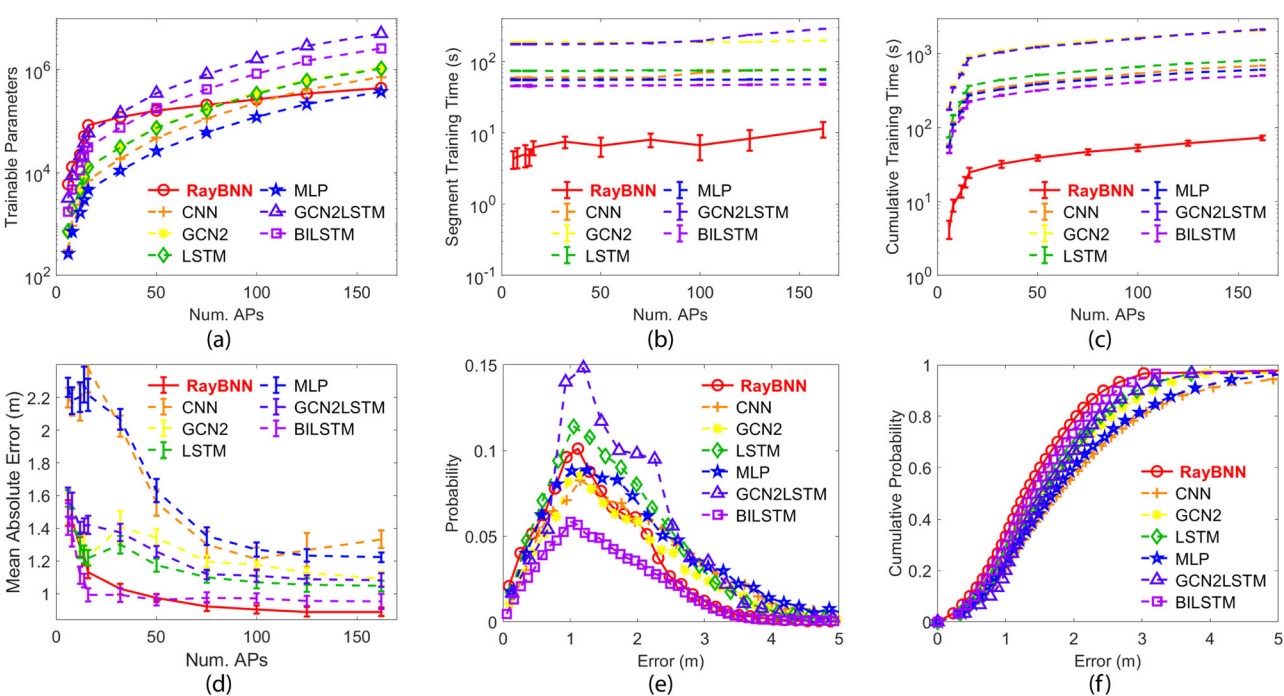

**Fig. 4 | Evolution of RayBNN on the Alcala Tutorial 2017 dataset with tenfold testing. a** MAE versus the RT-3 radius. **b** The probability density function of the values in the weighted adjacency matrix. **c** Absolute value percentile plot of the deleted weights. **d** The sparsity of the weighted adjacency matrix. **e** Plots of activation functions across different neurons. **f** Heat map of the weighted adjacency matrix.

Table 1 shows the algorithms' performances on a 210-GB EEG dataset[36]. In this dataset, there are 54 different subjects and each subject has two experimental sessions for classifying and detecting motor-imagery (MI) tasks, event-related potential (ERP), and steady-state visually evoked potential (SSVEP) tasks. Fifty-fourfold subject-independent testing is used to evaluate the models in Table 1. For each fold, one subject is selected for the testing dataset, while the other 53 subjects are selected for the training dataset to remove any overlap between the training dataset and the testing dataset. Moreover, there are no duplicate samples between the testing datasets in each fold. That way, the algorithms are evaluated on their ability to generalize across subjects. Accuracy, precision, recall, $F_1$ score, and area under

**Table 1 | Performances of the algorithms in the EEG motor-imagery dataset[36]**

| Model | Accuracy | Precision | Recall | $F_1$ score | ROC AUC |
|---|---|---|---|---|---|
| CSP-LDA[37,38] | 0.624 ± 0.092 | 0.638 ± 0.097 | 0.624 ± 0.092 | 0.609 ± 0.103 | 0.646 ± 0.121 |
| CSP-LR[37,38] | 0.625 ± 0.092 | 0.639 ± 0.097 | 0.625 ± 0.092 | 0.610 ± 0.103 | 0.646 ± 0.121 |
| Xdawn-MDM[39,40] | 0.712 ± 0.112 | 0.732 ± 0.106 | 0.712 ± 0.112 | 0.701 ± 0.123 | 0.770 ± 0.129 |
| Xdawn-LR[39,40] | 0.827 ± 0.087 | 0.835 ± 0.083 | 0.827 ± 0.087 | 0.826 ± 0.089 | 0.891 ± 0.083 |
| Deep4Net[41] | 0.836 ± 0.108 | 0.851 ± 0.094 | 0.836 ± 0.108 | 0.831 ± 0.121 | 0.914 ± 0.086 |
| Xdawn-Deep4Net-MLP | 0.836 ± 0.085 | 0.844 ± 0.081 | 0.836 ± 0.085 | 0.834 ± 0.086 | 0.920 ± 0.071 |
| Deep4Net-**RayBNN** | 0.846 ± 0.104 | 0.849 ± 0.103 | 0.846 ± 0.104 | 0.845 ± 0.104 | 0.906 ± 0.094 |
| Xdawn-Deep4Net-**RayBNN** | 0.856 ± 0.085 | 0.861 ± 0.082 | 0.856 ± 0.085 | 0.856 ± 0.086 | 0.926 ± 0.068 |

Fifty-fourfold testing. Subject-Independent. Confidence interval of $1\sigma$. The bold text represents the proposed model.

curve receiver operating characteristic (AUC ROC) are recorded for the various algorithms.

Common spatial pattern (CSP)[37,38] is widely used for extracting EEG features by decomposing the multivariate EEG signal into component eigenvalues and eigenvectors. After extracting the features, they are fed into linear discriminant analysis (LDA) or logistic regression (LR) for classification. As shown in Table 1, CSP-LDA is not very good at generalizing across different subjects for this specific dataset and has a very low mean accuracy of 62.4%. CSP-LR has a slightly better accuracy of 62.5%. On the other hand, researchers have used the Xdawn algorithm[39] from the pyRiemann python package[40] to extract features from EEG signals. Xdawn projects the high-dimensional Riemann manifold source space to the tangent space, which allows each class to be discerned more easily than the source space. Subsequently, the minimum distance to mean (MDM) algorithm is used to produce the final classification result. Each class has a centroid, and the data samples closest to a specific centroid will be assigned to that specific class. The combination of Xdawn and MDM (Xdawn-MDM) performs significantly better than CSP algorithms, as its accuracy of 71.2% is much higher. Furthermore, using Xdawn-LR increases the accuracy to 82.7%.

Deep4Net[41] was developed as the state of the art CNN model for classifying EEG signals, of which is made out of five blocks. Each block has a 2D convolutional layer, batch normalization layer, max pooling layer, and dropout layer. Moreover, the model does not have any fully connected layers but uses a logsoftmax function as its final layer. Deep4Net's 83.6% accuracy is higher than Xdawn's accuracy because the convolutional layers can denoise and extract more features than the Xdawn algorithm. To outperform the state of the art, we incorporate RayBNN together with Deep4Net, as shown in Fig. 5a. Since Deep4Net's final layer aggregates data and loses a lot of information, we extract outputs from Deep4Net's second last layer and feed it into RayBNN's input neurons. For RayBNN's architecture, there are 1400 input neurons, 1000 hidden neurons, and 600,000 neural connections. Subsequently, RayBNN produces the final classification result for the EEG dataset. For the Deep4Net-RayBNN combination, it has an accuracy of 84.6% which is higher than standalone Deep4Net and Xdawn-Deep4Net-MLP. As there is no optimal feature extraction algorithm for all subjects, we decided to create an ensemble of Xdawn-Deep4Net-RayBNN as shown in Fig. 5a. This is done by first training the Deep4Net-RayBNN combination and transferring the network to the Xdawn-Deep4Net-RayBNN ensemble. The transfer learning flexibility of RayBNN allows it to dynamically accept the 1400-element output from Deep4Net and the 990-element output from Xdawn to predict the final EEG classification result. For this specific case, the RayBNN has 2390 input neurons, 1000 hidden neurons, and 600,000 neural connections. Overall, the Xdawn-Deep4Net-RayBNN ensemble has the highest accuracy of 85.6%, with precision, recall, $F_1$ score, and AUC ROC being higher than the rest of the algorithms.

Figure 5b shows a comparison between the Xdawn-Deep4Net-RayBNN and its Xdawn-Deep4Net-MLP counterpart for one of the testing folds in the EEG dataset. The MLP has a dropout rate of 50% and the RayBNN has a sparsity of ~50%. As the number of trainable parameters increases, the ROC AUC also increases. However, the ROC AUC eventually reaches a limit, even though the number of trainable parameters keeps increasing. As shown in the figure, RayBNN performs much better than MLP due to having neural connection pruning and deleting redundant neurons. Figure 5c shows the performances of the algorithms on an individual subject basis. The Xdawn algorithm performs better for some subjects than the Deep4Net. Conversely, Deep4Net performs better for some subjects than the Xdawn algorithm. Due to the fact RayBNN uses both Xdawn and Deep4Net, it has the advantages of both and produces the highest accuracy for most of the test index. For the training time of the various algorithms, the CSP-LDA algorithm has a training time of 15.73 ± 0.91 s and CSP-LR has 15.51 ± 0.97 s. Moreover, Xdawn-MDM and Xdawn-LR have 19.21 ± 1.2 s and 19.05 ± 1.6 s, respectively. On the other hand, Deep4Net has a training time of 7271 ± 231 s, which is drastically higher. Subsequently, Xdawn-Deep4Net-MLP, Deep4Net-RayBNN, Xdawn-Deep4Net-RayBNN have 7324 ± 235 s and 7306 ± 233 s and 7326 ± 235 s, respectively due to the incorporation of Deep4Net.

Table 2 shows the statistical testing of each EEG algorithm in comparison to Xdawn-Deep4Net-RayBNN. The accuracy is calculated for each individual algorithm and fold. To compare, we select two algorithms and compute the difference in accuracy for each fold. We applied the paired t-test to the differences to get the $p$ values. The null hypothesis assumes the difference between the algorithms has a mean equal to zero. As all $p$ values are equal or less than $1.7968 \times 10^{-3}$, we reject the null hypothesis and assert the Xdawn-Deep4Net-RayBNN is statistically better than all of the other algorithms.

## Discussion

In this article, we randomly positioned neurons in a 3D sphere. As shown in Fig. 2e, the probability density function of the neuron lengths is a continuous Gaussian curve. This gives a lot of flexibility for creating many different neural connections and neural network structures. Alternatively, neurons may be arranged in a patterned fashion. For example, when neurons are arranged on a set of concentric sphere surfaces and only allow neural connections between neighboring surfaces, then the RayBNN topology becomes equivalent to a conventionally layered neural network. Overall, there are many possible periodic or chaotic arrangements for neurons and glial cells. It is possible that certain arrangements, along with certain connection rules, will lead to out-performance over the state of the art in a set of applications. It is also feasible to optimize the position of neurons and glial cells through training. Therefore, implementing them and exploring their characteristics will be exciting research in the future. In particular, one may study it with the knowledge transformed from

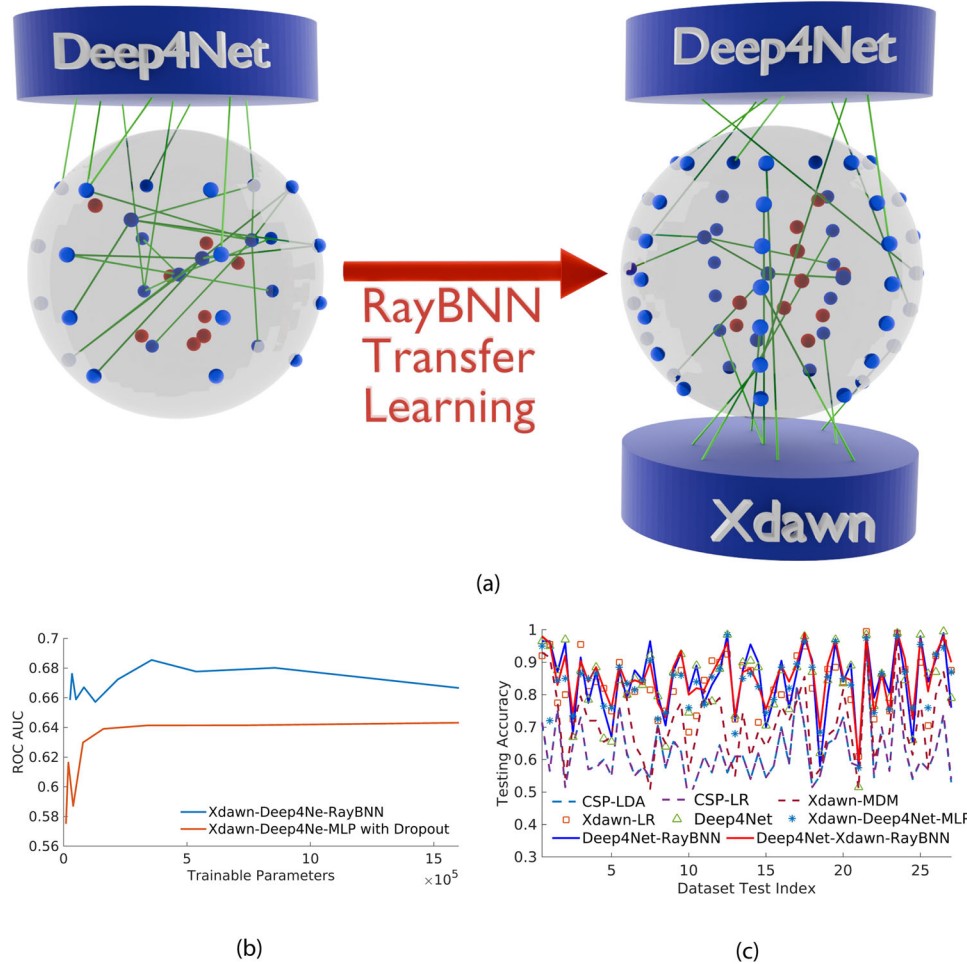

(a)

(b)

(c)

**Fig. 5 | Transfer learning algorithms on the EEG dataset. a** RayBNN transfer learning for EEG dataset and **b** Comparison of RayBNN and MLP in the EEG Dataset[36]. **c** EEG dataset and OpenBMI toolbox for three BCI paradigms: An Investigation into BCI Illiteracy[36]. Fifty-fourfold testing. Motor imagery.

group theory and solid state physics where various spatial topologies of atoms and molecules in solid have been extensively investigated.

The network's physical shape could be another exploration factor as it heavily influences the neural connection length and, in turn, also influences the propagation delay of information. For example, a neural network's overall shape could be a 3D cube, a torus, an ellipsoid, a tetrahedron, etc., each with its own advantages and disadvantages. In the case of a 3D cube or tetrahedron, the signal paths from the surface of the shape to the center are very different compared to a 3D sphere. As a result, some signals may reach the center of a cube faster, while others might take a longer time since they traveled a longer path. For applications needing synchronization, 3D cube and tetrahedron networks might not be optimal. For other applications requiring joining or merging neural networks, 3D cube, and tetrahedron networks can be easily joined together to significantly scale up a neural network for large datasets. In the case of an ellipsoid, the shape changes the number of hidden neurons that each input neuron can access, as it now becomes surface location dependent. Such bias toward a certain subset of input data features may be advantageous to train datasets whose data features are not equally important. It may be possible to evolve the network shape toward an ergodic optimal according to the nature of the dataset, just like how the human brain evolves from a sphere[42].

Overall, we created a three-dimensional RayBNN transfer learning model that is similar to real-life BNNs. In the world of machine learning, the traditional artificial neural network (ANN) is usually planar with well-structured neural network layers. Our RayBNN, like real-life BNN and unlike ANN, assigns 3D positions

to neurons and glial cells in a neural network sphere. The neurons are interconnected stochastically without well-defined layers, allowing information flow and learning transfers more efficient. Although still in its infant stage, our RayBNN has already out-performed conventional models in indoor localization, on both speed and accuracy. It also tops the state of the art in large EEG dataset analysis and predictions and demonstrates its capacity for seamless integration with conventional deep neural networks, which brings additional power to it. Note that up to date, the human brain still out-beats AI in many aspects, such as using symbolic logic to derive mathematical proofs, handling numerous incompatible data structures, and achieving multiple different objectives at the same time. We expect that with the continuing development, RayBNN will out-beat other AI models in these areas due to its inherent similarity to BNN.

As a human brain consumes much less power than current AI models, we intend to use real-life neurons, or in particular, optogenetically modified neurons, to implement our RayBNN so that the network can be trained and the input/output be read in/out optically, which may lead to a better AI hardware but with much lower power. Our research on RayBNN will not stop at developing better machine learning algorithms and hardware. The resemblance between the RayBNN and real-life biological neural networks makes RayBNN a unique platform for the studies of human and animal intelligence and behavior. With further development, the RayBNN neuromorphic device may be miniaturized so that it can be trained and implemented in patients for neural disease treatments.

## Methods

We display an overview of our RayBNN in Algorithm 1. Firstly, we assign 3D positions to glial cells and neurons in the "Cell location assignment and distribution analysis" section because they will form the physical structure of the neural network. As we randomly assign cell positions, some of those neurons and glial cells might intersect or clip into each other. We remove those intersecting cells following the methods and analysis in the "Cell collision detection and analysis" section. Secondly, new neural connections are ray-traced using the positions and radii of cells. "Forming neural connections via raytracing" Section lists the specialized raytracing algorithms for creating neural connections. Thirdly, every neural connection in the network is encoded into a sparse weighted adjacency matrix, as shown in "Mapping neural connections into the weighted adjacency matrix" Section. Meanwhile, details on implementing UAF to each neuron are discussed in the "Universal activation function" section. Subsequently, the forward pass uses the weighted adjacency matrix to calculate the neural network's output in the "RayBNN forward pass" section. In contrast, the backward pass produces the gradient of the weights, and the gradient descent algorithms apply it to update the weighted adjacency matrix in "Backpropagation" Section. During transfer learning, the dataset changes, which modifies the number of neurons and neural connections. If required, neural connections are deleted in "Deleting neural connections" Section and unused neurons are removed in "Deleting redundant neurons" Section.

**Algorithm 1.** Overview algorithm for RayBNN

---

**function** RayBNN($Model, \lambda, Dataset$):
 //Initialization
 Data $\leftarrow$ Dataset($\lambda$);
 Model($\lambda$).initNetworkSphere(Data);
 Model($\lambda$).raytraceConnections();
 Model($\lambda$).createWAdj();
 **while** *true* **do**
 //Training the Model
 Loss $\leftarrow \infty$;
 **while** *!isPlateau(Loss)* **do**
 Model($\lambda$).forwardPass(Data);
 Model($\lambda$).backwardPass(Data);
 Loss $\leftarrow$ Model($\lambda$).crossValidation(Data);
 **end while**
 //Transfer Learning
 Model($\lambda$+1) $\leftarrow$ Model($\lambda$);
 $\lambda \leftarrow \lambda + 1$;
 Data $\leftarrow$ Dataset($\lambda$);
 Model($\lambda$).delConnections();
 Model($\lambda$).delUnusedNeurons();
 Model($\lambda$).addNeurons();
 Model($\lambda$).raytraceConnections();
 **end while**

---

### Cell location assignment and distribution analysis

**Hidden neurons and glial cells' location assignment.** In our model, both hidden neurons and glial cells are uniformly distributed in a network sphere of radius $r_s$. To achieve that, we set up a spherical coordinate centered at the sphere origin with $(\hat{r},\hat{\theta},\hat{\phi})$ the unit vectors pointing to the radial, polar and azimuthal directions as shown in Fig. 6a. Within the sphere, every small volume $\delta V = r^2 \sin\theta \delta r \delta\theta\delta\phi$ centered at $(r, \theta, \phi)$ should contain the same number of cells, except

**Table 2 | Statistical testing of the EEG algorithms**

| Comparison | *p* value |
|---|---|
| Xdawn-Deep4Net-**RayBNN** vs Xdawn-Deep4Net-MLP | $1.2725 \times 10^{-4}$ |
| Xdawn-Deep4Net-**RayBNN** vs Deep4Net | $1.7968 \times 10^{-3}$ |
| Xdawn-Deep4Net-**RayBNN** vs Xdawn-LR | $2.2429 \times 10^{-6}$ |
| Xdawn-Deep4Net-**RayBNN** vs Xdawn-MDM | $1.0220 \times 10^{-16}$ |
| Xdawn-Deep4Net-**RayBNN** vs CSP-LR | $2.3756 \times 10^{-24}$ |
| Xdawn-Deep4Net-**RayBNN** vs CSP-LDA | $2.2413 \times 10^{-24}$ |

Paired *t*-test with right tail *p* values of the accuracy. Fifty-fourfold testing. The bold text represents the proposed model.

for statistical fluctuations. Therefore, to assign the location of a cell $i$, we first generate three random numbers $\mathcal{R}_r$, $\mathcal{R}_\theta$, and $\mathcal{R}_\phi$, each uniformly distributed within 0 to 1. Then the position of the cell $(r_i, \theta_i, \phi_i)$ can be assigned following the formula below:

$$r_i = \mathcal{R}_r^{1/3} r_s \tag{1}$$

$$\theta_i = \cos^{-1}[2 \times (\mathcal{R}_\theta - 0.5)] \tag{2}$$

$$\phi_i = 2\pi \times \mathcal{R}_\phi \tag{3}$$

To verify that the location assignment of cells is uniform within the sphere at a constant density $\eta_T = \frac{N_T}{r_s^3 4\pi/3}$ with $N_T = N_n + N_g$ being the total number of neuron ($N_n$) and glial ($N_g$) cells, we analyze the population density function of cells $n_T(r)$ on a sphere surface of radius $r$ and concentric to the network sphere, which is found to be

$$n_T(r) = \eta_T 4\pi r^2 = \frac{3N_T}{r_s^3} r^2. \tag{4}$$

The parabolic relation of the population distribution is confirmed in Fig. 2 as discussed in the previous section.

**Input and output neurons assignment.** In our model, we assign all output neurons to the center of the network sphere, while input neurons are at the surface of the sphere. In many cases, the features of input data are correlated and ordered. Therefore, the input neurons at the sphere surface should also maintain the same order and be equally spaced apart. For example, an image may contain $(N_x, N_y)$ pixels and their 2D order should not change. To accommodate that, we develop the input neuron assignment scheme as follows. We first create a $N_x \times 1$ vector $\vec{V}_\theta = [v_\theta^1, \ldots, v_\theta^i, \ldots, v_\theta^{N_x}]^T$ and a $N_y \times 1$ vector $\vec{V}_\phi = [v_\phi^1, \ldots, v_\phi^j, \ldots, v_\phi^{N_y}]^T$ such that all elements are equally spaced between 0 and 1

$$\begin{cases} v_\theta^i &= \frac{i+1}{N_x+1} \quad (i = 0, \ldots, N_x - 1) \\ v_\phi^j &= \frac{j+1}{N_y+1} \quad (j = 0, \ldots, N_y - 1) \end{cases} \tag{5}$$

Shown as the black dots in Fig. 1c, the location of the input neuron that corresponds to the $(i, j)$ pixel of the image can then map to the sphere according to

$$\begin{cases} r_{i,j} = r_s \\ \theta_{i,j} = cos^{-1}[2 \times (v_\theta^i - 0.5)] \\ \phi_{i,j} = 2\pi v_\phi^j \end{cases} \tag{6}$$

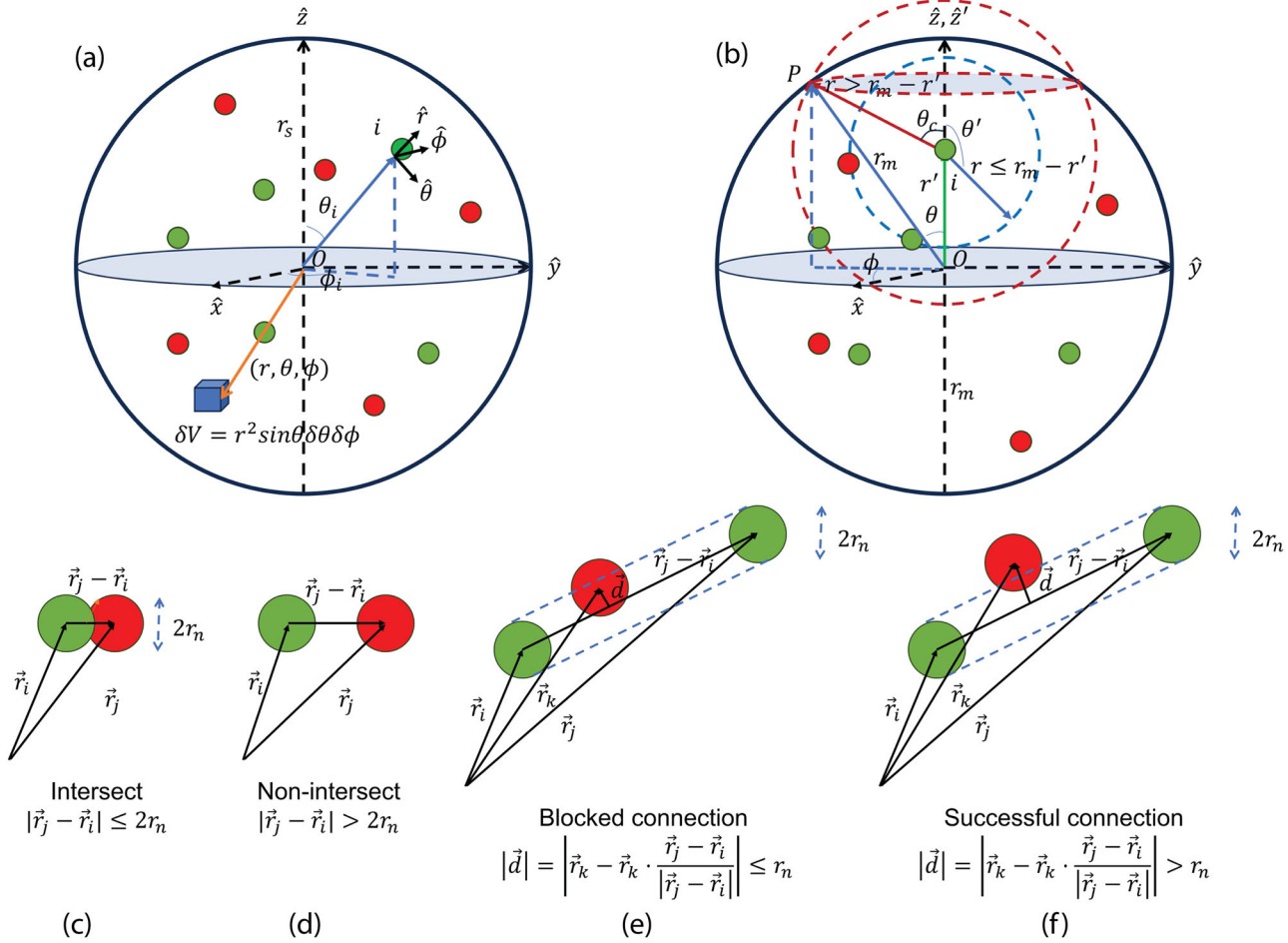

**Fig. 6 | Visualization of the ray-tracing method. a** Illustration of the global spherical coordinate $(\hat{r},\hat{\theta},\hat{\phi})$ centered at the origin of the network sphere with radius $r_s$. A small cube located at a position of $(r, \theta, \phi)$ has a differential volume of $\delta V = r^2 \sin\theta\delta\theta\delta\phi$. Both neurons (green balls) and glial cells (red balls) are uniformly distributed within the network sphere, leading to a parabolic cell density distribution along the radial direction. **b** Probability of neural connection calculation setup for RT-3. The origin $O$ of the cluster spherical coordinate $(\hat{r},\hat{\theta},\hat{\phi})$ is located at the center of the cluster sphere with a radius of $r_m$. A local spherical coordinate of neuron $i$ $(\hat{r}',\hat{\theta}',\hat{\phi}')$ is at the neuron center. Both coordinates are aligned so that $\hat{z}$ and $\hat{z}'$ are parallel to the line between $i$ and $O$. When a sub-cluster sphere centered at $i$ is within the cluster sphere ($r \le r_m - r'$, blue dashed sphere), all

neurons on that sub-cluster sphere surface may be accessible for neuron $i$ to form connections. If $r_m - r' < r \le r_m + r'$, the sub-cluster sphere intersects the cluster sphere (red dashed sphere). Only neurons on the sub-cluster sphere surface within the cluster sphere are accessible by neuron $i$. When $r > r_m + r'$, the sub-cluster surface is outside the cluster sphere and none of the neurons on its surface are accessible by neuron $i$. **c** Two neurons $i$ and $j$ intersect if their distance $|\vec{r}_j - \vec{r}_i| \le 2r_n$. **d** Neurons do not intersect if $|\vec{r}_j - \vec{r}_i| > 2r_n$. **e** Neurons $i$ and $j$ can not form a connection if the distance of a third cell $k$ to the connection $|\vec{d}| \le r_n$. **f** A connection will be formed if $|\vec{d}| \le r_n$. Note that cell $k$ must be in-between neurons $i$ and $j$, or $(\vec{r}_k - \vec{r}_i) \cdot (\vec{r}_j - \vec{r}_i) > 0$ and $(\vec{r}_k - \vec{r}_j) \cdot (\vec{r}_i - \vec{r}_j) > 0$.

---

Note that since each input neuron occupies the same solid angle $\delta\Omega = \frac{4\pi}{N_x N_y}$ and thus the same area of the sphere surface, it will have unbiased access to the hidden neurons as they are uniformly distributed under the surface. Meanwhile, the order and correlation of the pixels in the original image are preserved. Moreover, the input neurons can be easily mapped to 1D data so that the order of the needed features are preserved. For example, to map a 1D, $N_{1D}$-point EEG data to our 2D sphere surface, we should build $\vec{V}_\theta$ and $\vec{V}_\phi$ with $N_x = N_y = \lceil \sqrt{N_{1D}} \rceil$ and then flatten the 2D neuron location into a 1D vector $\vec{A}$ with a helix pattern $\vec{A} = [(0,0),(0,1),\ldots,(0,N_y - 1),(1,0), (1,1),\ldots,(1,N_y - 1),\ldots]^T$ showing as the blue line in Fig. 1c.

To map 3D ordered data such as RGB images to the input neurons, we can assign red, blue, and green pixels to the same location. When a hidden neuron tries to create a connection to the location where the three neurons are, it will randomly pick one of them to connect.

The input neuron location assignment can be further simplified if the input features are not ordered. In this case, the neurons can be randomly assigned on the surface with the exact method to assign hidden neurons except that their radial coordinates will be fixed at $r_s$.

## Cell location re-assignment upon population growth

When transferring knowledge between datasets, the dimensions of the datasets might change. This is reflected in the number of input and output neurons. If the dimension increases, then more neurons are added to the input and output neurons. In addition, the number of hidden neurons and glial cells may also increase to accommodate the increasing complexity of the new dataset. In this case, the network sphere will increase to $r_s'$ to retain the low collision rate. To achieve that, we first relocate all of the old cells to the new network sphere by simply changing their radial position to $r_i' = \frac{r_s'}{r_s} r_i$ while keeping the polar and azimuthal angles fixed. The new cells are then added to the expanded sphere using the same procedure described above. Similarly, one may increase the input neurons on the sphere surface in an ordered pattern depending on the way the new dataset is formed. For example, if the dataset is transferred from low-resolution images to higher resolution, one may simply densify $\vec{V}_\theta$ and $\vec{V}_\phi$ by inserting new elements evenly within each vector. In this way, new neurons shown as the red dots in Fig. 1d can be located according to the new vector elements while old input neurons can stay at their original locations

without the need to reconnect. On the other hand, if the new dataset concatenates new features to the previous dataset, then the old neurons can simply move toward, eg. north of the sphere as shown in Fig. 1e by recalculating $\theta_i = cos^{-1}[2 \times (\kappa v_\theta^i - 0.5)]$ with the connections to hidden neurons retained. Here $0 < \kappa \le 1$ is a densification factor that determines how much space in the south that needs to be emptied for the new neurons. Meanwhile, the new input neurons can be added, e.g., on the south of the sphere in the space emptied from the old neurons.

**Cell collision detection and analysis.** During the cell location assignment, some cells may collide. In our model, we delete all colliding cells during the assignment. As deleting cells is computationally costly, we keep the collision rate below 1%. This requires that the network sphere radius must be larger than the minimum radius $r_{s,min}$ to keep the cell density sparse. As shown in Fig. 6c, d, a collision occurs to a cell at $r_i$ if the center of another cell is within $2r_n$ distance. Further, cells are uniformly distributed within the sphere and $r_s \gg 2r_n$. Therefore, in a new spherical coordinate $(\hat{r}', \hat{\theta}', \hat{\phi}')$ centered at cell $i$, neglecting the cells that are within $2r_n$ of the network sphere surface, we may expect the population density function at $r'$ has the same form as Eq. (4), $n_T(r') = \frac{3N_T}{r_s^3} r'^2$. Therefore, the collision probability can be written as

$$P_c = \int_0^{2r_n} n_T(r') dr' = 8N_T \frac{r_n^3}{r_s^3} = \frac{32\pi}{3} \eta_T r_n^3 \quad (7)$$

as long as $P_c \ll 1$. Therefore, at a preset minimum collision threshold $P_{c,th}$, the cell density must satisfy

$$\eta_T < \frac{3P_{c,th}}{32\pi r_n^3} \quad (8)$$

while the sphere radius

$$r_s > \left(\frac{8N_T}{P_{c,th}}\right)^{\frac{1}{3}} r_n \quad (9)$$

Eq. (7) can also be explained as follows. In a network sphere of radius $r_s$ and volume $V_s = \frac{4\pi}{3} r_s^3$, if the density of cells is sufficiently sparse so that the number of cells that intersect each other is much fewer than the total number of cells. Cell intersection occurs only when a cell falls within the volume $V_n = \frac{4\pi}{3}(2r_n)^3$ occupied by any other cell. Therefore, the probability to place a single cell into the network sphere and intersect with any other cells is $P_c = N_T V_n / V_s = \frac{32\pi}{3} \eta_T r_n^3$. Since there are $N_T$ cells, the total number of intersect cells will be $N_c = N_T P_c$, resulting in the collision rate $\frac{N_c}{N_T} = P_c$, which is consistent with Eq. (7).

**Forming neural connections via raytracing**
We implemented three different raytracing (RT) algorithms for connecting neurons together. In RT algorithm 1 (RT-1): randomly generated rays, each neuron randomly outputs $K$ rays of random angles and of infinite lengths. Typically, $K$ should be larger than the number of connections each neuron would make. In our case, we set $K = 10,000$ to ensure sufficient neural connections. For a network of $N_n$ neurons, there are $KN_n$ randomly generated rays. If a ray intersects a glial cell, then it is removed. If a ray intersects multiple neurons, then one new neural connection is created from the current neuron to the closest intersected neuron, while the neurons past it are not connected. The algorithm for detecting the intersection is as follows. It generates rays of random lengths and directions. Subsequently, our algorithm checks the generated rays to see if they intersect any other cells, or equivalently, if there is a cell's distance to any ray is within $r_n$. If a

ray intersects a neuron and not a glial cell, then the ray is inserted into a queue. Meanwhile, duplicate neural connections occupying the same space are removed from the queue. In total, RT-1 requires $KN_n(N_n + N_g)$ comparisons, and it is inefficient because some rays intersect the same object multiple times and other rays do not intersect anything. Duplicates of the same connections are removed using a deduplication algorithm.

To make the algorithm more efficient, we created RT algorithm 2 (RT-2): directly connected rays, where each neuron is directly connected to every other neuron in the neural network via a finite-length ray. Thus, $N_n^2$ rays are generated, and they are compared to $N_n + N_g$ neurons and glial cells. Again, rays that intersect glial cells are removed and rays that intersect multiple neurons will end at the closest neuron. RT-2 also uses the same ray intersection algorithm and deduplication algorithm. In total, there are $N_n^2(N_n + N_g)$ comparisons, which is inefficient for large sizes of neurons as the complexity increases to $O(N_n^3)$.

Building upon the previous algorithm and assuming far-reaching connections can be ignored, we propose RT algorithm 3 (RT-3): distance-limited directly connected rays. Firstly, a random cell is selected as a pivot. A segment is constructed by only selecting cells within a fixed sphere radius ($r_m$) of the pivot, which has approximately $N_m$ neurons and $N_{gm}$ glial cells. Afterwards, the RT-2 is applied to the segment to generate new neural connections and the process repeats by selecting new pivots. New neural connections from each segment are concatenated and deduplicated to remove multiples of the same connection. Each segment has $N_m^2$ rays that are compared to $N_m + N_{gm}$ cells, therefore there are $N_m^2(N_m + N_{gm})$ comparisons per segment. Assuming the network is divided into $K$ segments, the total number of comparisons is approximately $KN_m^2(N_m + N_{gm})$. As the total number of neurons is much greater than the number of neurons in a segment $N_n \gg N_m$, this speeds up RT-3 by a factor of $(N_n)^2/KN_m^2 \gg 1$ over RT-2. We also ensure all output neurons are connected to all input neurons by traversing the network backward and checking all neural connections.

**Neural connection length probability distribution function.** In this subsection, we derive the neural connection length probability using RT-3. Here, for simplicity, we assume each cluster in RT-3 is spherical in shape with radius $r_m$. As shown in Fig. 6b, we also adopt a cluster spherical coordinate $(\hat{r}, \hat{\theta}, \hat{\phi})$ whose origin is at the center of the cluster sphere (O) and a local spherical coordinate $(\hat{r}', \hat{\theta}', \hat{\phi}')$ whose origin is at a neuron $i$ that is $r' < r_m$ away from the cluster center. Further, we align the $\hat{z}$ axis of both coordinates such that the position of neuron $i$ can be written as $(r', 0, 0)$ in the cluster coordinate. As shown in Fig. 6e, f, a connection between neuron $i$ and another neuron $j$ that is $r$ distance away will not form if there is a cell $k$ to block the line of sight. Therefore, if the cell density is sufficiently sparse, the probability of not forming a connection should equal the number of cells in the cylinder that connects these two neurons and have a circular cross-section of radius $r_n$, leading to the probability of making a successful connection to be

$$p(r) = 1 - \pi r_n^2 \eta_T r \quad (10)$$

Therefore, the conditional probability of neuron $i$ forming a connection of length $r$ is

$$P_{nc}(r|r') = n_n(r) \times p \quad (11)$$

Where $n_n(r) = n_T(r)$, the population density of neurons $r$ distance away from neuron $i$ is half of the total population density as we have equal numbers of neurons and glial cells. Note that for $r < r_m - r'$ (blue dashed sphere in Fig. 6b), all connecting neurons are within the cluster.

Following the derivation in the previous section.

$$n_{nc}(r|r') = 4\pi\eta_n r^2(1 - \pi r_n^2 \eta_T r) \tag{12}$$

On the other hand, if $r_m - r' < r < r_m + r'$ (red dashed sphere in Fig. 6b), only a portion of neurons have the same distance $r$ are inside the cluster while those outside can not make connections. The portion of the qualified neuron can be estimated using the solid angle of the crust that is inside the cluster, from which we get

$$n_{nc}(r|r') = \frac{2\pi \int_{\theta_c}^{\pi} \sin\theta' d\theta'}{4\pi} 4\pi\eta_n r^2(1 - \pi r_n^2 \eta_T r) \tag{13}$$

$$= \pi\eta_n \frac{r[r_m^2 - (r - r')^2](1 - \pi r_n^2 \eta_T r)}{r'} \tag{14}$$

Finally, $N_c(r|r') = 0$ for $r \geq r_m + r'$ since no qualified neurons are in the cluster. Further, following the Bayesian Theorem, we have the connection length probability

$$
\begin{aligned}
P_{nc}(r) &= \int_0^{r_m} dr' P_{nc}(r|r') P_n(r') \\
&= K(1 - \pi r_n^2 \eta_T r) \left\{ \int_0^{r_m-r} dr' r^2 r'^2 + \int_{r_m-r}^{r_m} dr' \frac{rr'[r_m^2 - (r-r')^2]}{4} \right\} \\
&= Kr^2(1 - \pi r_n^2 \eta_T r)(r - 2r_m)^2(r + 4r_m)
\end{aligned}
\tag{15}
$$

where $P_{nc}(r|r') = \frac{n_{nc}}{N_{nc}}$ is the probability density of forming a connection of length $r$ condition of the neuron $i$ is at $r'$ and $P_n(r') = \frac{n_n(r')}{N_n}$ being the probability of neuron $i$ is at $r'$. $N_{nc}$ is the total number of connections within the cluster, $K$ is a normalization factor such that $\int_0^{2r_m} P_{nc}(r) dr = 1$. Note that, in our case $\eta_T$ is sufficiently small such that $P_{nc}(r) \approx Kr^2(r - 2r_m)^2(r + 4r_m)$ which is independent to the density. The density independence at low density was confirmed in Fig. 2e.

## Mapping neural connections into the weighted adjacency matrix

After generating the neural connections via the raytracing algorithms, they are mapped into the $N \times N$ weighted adjacency matrix $\mathbf{W}$. The total neuron capacity $N \in \mathbb{N}^+$ controls how many neural connections are reserved in memory. In every case, the neuron capacity is greater than the number of neurons $N > N_n^\lambda, \forall\lambda$ to allow adding/deleting neurons without resizing/reallocating the weighted adjacency matrix $\mathbf{W}$. Here, the superscript $\lambda$ stands for the $\lambda$-th evolution of transfer learning. Each individual matrix element $\{w_{ij}, \{i, j\} \in 1, ..., N\}$ represents the weight of a unidirectional ray-traced connection from $i$-th neuron to $j$th neuron. Following that, the weights $w_{ij}$ are initialized with Xavier weight initialization algorithm[43].

Storing the entire weighted adjacency matrix together with the zero element weights takes too much memory space and is computationally expensive for matrix multiplication. To solve this problem, $\mathbf{W}$ is stored in compressed sparse row (CSR) matrix format, where the value vector $\vec{W}_{value}$ only stores the non-zero elements. While CSR matrices are used for computing forward pass and backward pass as mentioned in "RayBNN Forward Pass" and "Backpropagation" subsections, we use COOrdinate format (COO) matrices to add and delete new weights/neural connections. More information on the sparse matrix format can be found in Supplementary Materials S.3.A[25] and S.3.B[25].

Some neural connections hamper the performances of the neural networks by overriding certain values and network states. For example, input neurons that connect to other input neurons hamper the data flow because they override current input values with the previous

input values from the previous time step. Deleting these values from the weighted adjacency matrix $\mathbf{W}(0:N_I-1,:) = 0$ severs the neural connections from the input neurons to themselves and fixes the problem. Similar to the problem above, output neurons that connect to other output neurons produce incorrect neural network outputs because they override the current output values. Again, the problem can be solved by deleting weights connecting output neurons to other output neurons $\mathbf{W}(:, N - N_O - 1 : N - 1) = 0$.

When a neuron connects to itself, this is called a self-loop[44]. For datasets requiring memory-less neural networks, self-loops may degrade the performance of the neural network because there could be a positive feedback cycle that goes to positive infinity. Self-loops can be removed by setting the diagonals of the weighted adjacency matrix to zero $diag(\mathbf{W}) = 0$.

## Universal activation function

There are many different activation functions in machine learning, and it is difficult to determine the optimal activation function for a certain application. To solve the problem, we adopted the universal activation function (UAF)[45] to dynamically evolve the UAF to the best activation function. An example of UAF is presented in Supplementary Information, Section S.4[25]. Here, we apply a unique UAF to the output of every neuron in the network by modifying the single input single output version of the UAF to a multiple input multiple output version of the UAF. After the modification, each neuron in the network has five unique parameters that specifically control its own specific UAF.

For example, the gradient descent algorithm could tune the parameters such that the UAF evolves to the LeakyReLU function for some neurons, while evolving to the Tanh function for other neurons. The single input, single output version of the UAF $f_{\text{UAF}}(x)$

$$f_{\text{UAF}}(x) = \ln(1 + e^{A(x+B) - |C|x^2}) - \ln(1 + e^{D(x-B)}) + E \tag{16}$$

takes in an input $\forall x \in \mathbb{R}$ and produces an output based on the trainable parameters $\forall A, B, C, D, E \in \mathbb{R}$.

In this article, we further extend the UAF to multiple input/multiple output cases.

$$\vec{X} = \begin{bmatrix} x_0 \\ x_1 \\ x_2 \\ \vdots \\ x_{N-1} \end{bmatrix}, \hat{Y} = \begin{bmatrix} y_0 \\ y_1 \\ y_2 \\ \vdots \\ y_{N-1} \end{bmatrix} \tag{17}$$

$$\mathbf{C_{eff}} = \begin{bmatrix} A_0 & B_0 & C_0 & D_0 & E_0 \\ A_1 & B_1 & C_1 & D_1 & E_1 \\ A_2 & B_2 & C_2 & D_2 & E_2 \\ \vdots & \vdots & \vdots & \vdots & \vdots \\ A_{N-1} & B_{N-1} & C_{N-1} & D_{N-1} & E_{N-1} \end{bmatrix} \tag{18}$$

$$y_i = f_{\text{UAF}}(x_i, A_i, B_i, C_i, D_i, E_i) \tag{19}$$

$$\hat{Y} = \vec{f}_{\text{UAF}}(\vec{X}, \mathbf{C_{eff}}) \tag{20}$$

where $N$ is the length of the input vector $\vec{X} \in \mathbb{R}^{N \times 1}$ and the output vector $\hat{Y} \in \mathbb{R}^{N \times 1}$. $\mathbf{C_{eff}} \in \mathbb{R}^{N \times 5}$ is a matrix filled with coefficients $\forall A_i, B_i, C_i, D_i, E_i \in \mathbb{R}$ that describes the shapes of the individual activation functions. $\vec{f}_{\text{UAF}}$ is applied element-wise to the input vector $\vec{X}$ that contains input variables $\forall x_i \in \mathbb{R}$ and produces the output vector $\hat{Y}$ that contains the output values $\forall y_i \in \mathbb{R}$.

## RayBNN forward pass

When the weighted adjacency matrix is finally configured, we want to get the output states of the neural network at every time step $t$. The neural network contains many external and internal states, of which record the output values of individual neurons. The input state vector contains information that will be placed into the input neurons, while the output state vector contains information extracted from the output neurons. On the other hand, the internal state vector keeps track of every single active neuron inside the neural network. At every time step $t$, the forward pass algorithm places the input state vector into the input neurons. Simultaneously, the algorithm updates the current internal state vector using the previous internal state vector and the input state vector, while extracting the output state vector from the output neurons.

Now for the mathematical description of the forward pass algorithm. The input state vector $\vec{X}^t \in \mathbb{R}^{N_I \times 1}$ and the output state vector $\hat{Y}^t \in \mathbb{R}^{N_O \times 1}$ are created

$$
\vec{X}^t = \begin{bmatrix} x_0^t \\ x_1^t \\ x_2^t \\ \vdots \\ x_{N_I-1}^t \end{bmatrix}, \hat{Y}^t = \begin{bmatrix} y_0^t \\ y_1^t \\ y_2^t \\ \vdots \\ y_{N_O-1}^t \end{bmatrix} \tag{21}
$$

with $N_I$ number of input elements and $N_O$ number of output elements respectively. Note that each input $\vec{X}^t$ is synchronized with output $\hat{Y}^t$ for training purposes. Meanwhile, the neuron bias vector $\vec{H} \in \mathbb{R}^{N \times 1}$ and the internal state vector $\vec{S}^t \in \mathbb{R}^{N \times 1}$ are initialized

$$
\vec{H} = \begin{bmatrix} h_0 \\ h_1 \\ h_2 \\ \vdots \\ h_{N-1} \end{bmatrix}, \vec{S}^t = \begin{bmatrix} s_0^t \\ s_1^t \\ s_2^t \\ \vdots \\ s_{N-1}^t \end{bmatrix} \tag{22}
$$

to have the same neuron size $N$. Typically, the bias vector $\vec{H}$ is initialized with random normal numbers that are later trained by the gradient descent algorithms. However, the state vector at time $t = 0$ is always initialized with all zero elements $\vec{S}^0 = \vec{0}$ to ensure the initial neuron state is blank.

At every time step $t$, a temporary state vector $\vec{Q}^t \in \mathbb{R}^{N \times 1}$

$$
\vec{Q}^t \leftarrow \vec{S}^t \tag{23}
$$

is created using the current state vector $\vec{S}^t$. Following that, the input vector $\vec{X}^t$ is placed into elements index 0 to index $N_I - 1$ of the temporary state vector $\vec{Q}^t$

$$
\vec{Q}^t(0 : N_I - 1) \leftarrow \vec{X}^t \tag{24}
$$

so that the input neurons' values are updated with the current input vector. As our objective is to propagate the input information throughout the hidden neurons, we update the state of every neuron that is directly connected to the current set of neurons. This is done by computing the next state vector $\vec{S}^{t+1}$

$$
\vec{S}^{t+1} \leftarrow \vec{f}_{UAF}(\mathbf{W}\vec{Q}^t + \vec{H}, \mathbf{C}_{eff}) \tag{25}
$$

where the weighted adjacency matrix $\mathbf{W}$ multiplies the temporary state vector $\vec{Q}^t$. Afterward, the bias vector $\vec{H}$ is added to the resulting vector, and the result goes through the activation function $\vec{f}_{UAF}$.

In order to ensure the input information reaches the output neurons, the process above is repeated $U$ total time steps to yield a time sequence of state vectors $\{\vec{S}^0, \vec{S}^1, \vec{S}^2, \ldots, \vec{S}^{U-1}\}$. $U = I_T + k$ is the total number of processing steps, where $I_T$ is the number of input vectors in the time series and $k$ is the programmed propagation delay between sending an input vector and receiving an output vector. Typically, the value $k$ is greater or equal to the mean traversal depth from the input neurons to the output neurons. Higher values of $k$ allow the neural network to perform more complex computational tasks at the cost of more computational time and larger memory usage. Now, the output vectors are extracted from the output neurons. For example, an output vector $\hat{Y}^t$ at time $t$ is constructed using elements index $N - N_O$ to index $N - 1$ of the state vector $\vec{S}^{t+k}$ at time $t + k$

$$
\hat{Y}^t \leftarrow \vec{S}^{t+k}(N - N_O : N - 1) \tag{26}
$$

where each output vector $\hat{Y}^t$ corresponds to an input vector $\hat{X}^t$.

For a simple example, imagine a state vector $\vec{S}^0$ that has all zeros

$$
\vec{S}^0 = \begin{bmatrix} 0.0 \\ 0.0 \\ 0.0 \\ 0.0 \end{bmatrix} \tag{27}
$$

and is used to update $\vec{Q}^0 \leftarrow \vec{S}^0$ together with input vector $\vec{Q}^t(0:1) \leftarrow \vec{X}^0$.

$$
\vec{X}^0 = \begin{bmatrix} -2.1 \\ 0.3 \end{bmatrix}, \vec{Q}^0 = \begin{bmatrix} -2.1 \\ 0.3 \\ 0.0 \\ 0.0 \end{bmatrix} \tag{28}
$$

The next state vector $\vec{S}^1$ is computed using $\vec{Q}^0$ and $W$ assuming the UAF is an identity function and $\vec{H}$ is all zeros.

$$
\mathbf{W} = \begin{bmatrix} 0.2 & 1.5 & -0.1 & 0.9 \\ -0.8 & 0.0 & 0.6 & -1.0 \\ 1.0 & -0.7 & 3.2 & 1.2 \\ -2.0 & -0.5 & 0.7 & 0.1 \end{bmatrix} \tag{29}
$$

$$
\vec{S}^1 = \begin{bmatrix} 0.2 & 1.5 & -0.1 & 0.9 \\ -0.8 & 0.0 & 0.6 & -1.0 \\ 1.0 & -0.7 & 3.2 & 1.2 \\ -2.0 & -0.5 & 0.7 & 0.1 \end{bmatrix} \begin{bmatrix} -2.1 \\ 0.3 \\ 0.0 \\ 0.0 \end{bmatrix} \tag{30}
$$

Following that, the output vector $\vec{Y}^0$ is extracted from the state vector $\vec{S}^1$.

$$
\vec{S}^1 = \begin{bmatrix} 0.03 \\ 1.68 \\ -2.31 \\ 4.05 \end{bmatrix}, \vec{Y}^0 = \begin{bmatrix} -2.31 \\ 4.05 \end{bmatrix} \tag{31}
$$

## Backpropagation

We use gradient descent algorithms to optimize the parameters of the RayBNN. However, they require the gradients of the weights and biases. A modified backpropagation algorithm, made specifically for CSR matrices, is used to compute those gradients. Firstly, the overall loss function $J$ is computed

$$
J = \frac{1}{I_T} \sum_{t=0}^{I_T} L(\hat{Y}^t, \vec{Y}^t) \tag{32}
$$

using the loss function $L$, the neural network prediction $\hat{Y}$, and the actual output $\vec{Y}$. Secondly, the CSR-weighted adjacency matrix **W** is flattened into a 1D weight vector $\vec{W}$, where the elements are in row-major order. This allows us to update certain elements in the weighted adjacency matrix **W** without updating all elements. We want to find the gradient of the loss function with respect to the weights $\frac{\partial J}{\partial \vec{W}}$ by evaluating the partial derivative of the loss function $\frac{\partial L}{\partial \hat{Y}^t}$ at $(\hat{Y}^t, \vec{Y}^t)$

$$\frac{\partial J}{\partial \vec{W}} = \frac{1}{I_T} \sum_{t=0}^{I_T} \left[ \frac{\partial L}{\partial \hat{Y}^t} \bigg|_{(\hat{Y}^t, \vec{Y}^t)} \odot \frac{\partial \hat{Y}^t}{\partial \vec{W}} \right] \qquad (33)$$

evaluating the partial derivative of the activation function $\vec{f}_{UAF}$ with respect to input vector $\vec{X}$. Note that $\odot$ represents element-wise tensor/matrix/vector multiplication. Moreover, the partial derivatives $\frac{\partial L}{\partial \hat{Y}^t}, \frac{\partial \hat{Q}}{\partial \vec{W}}, \frac{\partial \vec{f}_{UAF}}{\partial \vec{X}}$ are reshaped to match the dimensions of $\vec{W}$.

$$\frac{\partial \hat{Y}^t}{\partial \vec{W}} = \frac{\partial \vec{f}_{UAF}}{\partial \vec{X}} \bigg|_{\mathbf{W}\vec{Q}^{t+k-1} + \vec{H}} \odot \frac{\partial}{\partial \vec{W}} \left[ \mathbf{W}\vec{Q}^{t+k-1} \right] \qquad (34)$$

Partial derivative of the state vector $\frac{\partial \vec{Q}}{\partial \vec{W}}$ is recursively computed until $\frac{\partial \vec{Q}}{\partial \vec{W}}^0$ is reached.

$$\frac{\partial}{\partial \vec{W}} \left[ \mathbf{W}\vec{Q}^{t+k-1} \right] = \vec{Q}^{t+k-1} + \mathbf{W}\frac{\partial \vec{Q}}{\partial \vec{W}}^{t+k-1} \qquad (35)$$

$$\frac{\partial \vec{Q}}{\partial \vec{W}}^{t+k-1} = \frac{\partial \vec{f}_{UAF}}{\partial \vec{X}} \bigg|_{\mathbf{W}\vec{Q}^{t+k-2} + \vec{H}} \odot \frac{\partial}{\partial \vec{W}} \left[ \mathbf{W}\vec{Q}^{t+k-2} \right] \qquad (36)$$

For a simple example, assume an MSE loss function and the equations from the previous subsection

$$J = (\hat{Y}^0 - \vec{Y}^0)^{\mathsf{T}} (\hat{Y}^0 - \vec{Y}^0) \qquad (37)$$

$$\frac{\partial J}{\partial \vec{W}} = 2(\hat{Y}^0 - \vec{Y}^0) \odot \frac{\partial \hat{Y}^0}{\partial \vec{W}} \qquad (38)$$

$$\vec{f}_{UAF} = \vec{X}, \frac{\partial \vec{f}_{UAF}}{\partial \vec{X}} \big|_{\mathbf{W}\vec{Q}^0 + \vec{H}} = \vec{1} \qquad (39)$$

$$\frac{\partial \hat{Y}^0}{\partial \vec{W}} = \vec{1} \odot \left[ \vec{Q}^0 + \mathbf{W}\frac{\partial \vec{Q}}{\partial \vec{W}}^0 \right] \qquad (40)$$

### Deleting neural connections
When transferring between datasets, the dataset size and the number of inputs could decrease. This might create overfitting, resulting in lower performance. The problem can be alleviated by reducing the amount of trainable parameters via deleting neural connections. The smallest weights of the neural network have the least effect on the output of the neural network, and thus they are deleted by finding the indexes of the 5% smallest weights $\vec{i},\vec{j}$ and setting those elements $\mathbf{W}(\vec{i},\vec{j})$ to zero. COO matrix element deletion is described in the Supplementary Material Subsection S.3.A[25]. Deleting too many neural connections could cause the network to get stuck at a local minimum. Repeatedly adding and deleting neural connections at every epoch could cause the loss function to oscillate out of the local minimum and to descend into the global minimum.

**Algorithm 2**. Deleting neural connections
 $\mathbf{R} \leftarrow$ randomUniform(0.0,1.0);
 $\mathbf{R} \leftarrow$ elemwiseMulti($\mathbf{R},\mathbf{W}$);
 $\vec{i},\vec{j} \leftarrow$ argmin($|\mathbf{R}|$);
 $\mathbf{W}(\vec{i},\vec{j}) \leftarrow 0.0$;
 To overcome the problem, we probabilistically delete weights such that larger weights still can be removed, but at a much lower probability than the smaller weights. This is implemented in Algorithm 2, where a random matrix **R** with the same dimensions as the weighted adjacency matrix **W** is initialized with random uniform numbers between 0.0 and 1.0. Then, the random matrix element-wise multiplies with the weighted adjacency matrix and the result is saved into the random matrix. Elements in the weighted adjacency matrix are set to zero based on the indexes of the 5% smallest values in the random matrix.

### Deleting redundant neurons
When deleting neural connections, non-zero elements in the weighted adjacency matrix **W** are deleted. However, some neurons are rendered redundant because they have all of their outputs removed but still have input weights. They can be safely deleted without affecting the neural network's output and performance. Useless neurons are detected by looking at the output degrees $D_{\text{out}}(P_i)$ of neurons $P_i$ and seeing which neurons have no outputs $D_{\text{out}}(P_i) = 0$. Subsequently, we check to determine there are any input neural connections to the useless neurons. If the input degrees are greater than zero, then the neurons' input neural connections/weights are removed by setting the elements in row $i$ to zero $w_{i,0} = w_{i,1} = w_{i,2} = ... = w_{i,N-1} = 0$. Moreover, deleted neurons have their cell positions $\vec{P}_i$ removed from the master list of all neuron positions.

The difference between deleting redundant neurons and dropout is that dropout randomly deletes neurons and neural connections, of which changes the outputs of the neural network/layer. On the other hand, deleting redundant neurons does not change the outputs of the neural network because the redundant neurons are not outputting any information into other neurons.

### Details of other models for comparison
For the CNN model, we used one CNN layer and two dense layers. The CNN layer has 4 channels, and each channel has a $5 \times 1$ convolutional filter. Each dense layer contains twice the amount of neurons as the input size. The hidden layers use the LeakyReLU activation function, while the final layer used the identity activation function. For the MLP model, we used three dense layers, where the number of neurons in each layer equals the input size. For the LSTM model, we used two LSTM layers and one dense layer for the final layer. Each LSTM layer has the same number of neurons as the input size. Moreover, the LSTM layers use the tanh activation function, while the dense layer used the identity activation function. Similar to the LSTM model above, the BiLSTM model has the exact same structure, except the LSTM layers are replaced with BiLSTM layers and the dense layers are twice the size to match the BiLSTM layers. For the GCN2 model, it used four graph convolutional layers and two linear graph layers. As an analog to CNN layers, the graph convolutional layers perform convolutions on the input nodes and edges to predict output nodes. Subsequently, the predicted result is formatted by the linear graph layers. Similar to the GCN2 model, the GCN2LSTM model has the exact same structure, but the graph convolutional layers are replaced with graph LSTM layers. Furthermore, we used tenfold testing to ensure reproducibility. In each fold, the networks are initialized with random weights and are trained accordingly. Afterward, we test each fold independently to get the mean absolute error (MAE) and training time standard deviation.

### Software framework
We have created an entirely custom software framework for simulating physical neurons and training biological neural networks. The Rust

programming language is chosen because it has compile-time code verification to prevent data races, array indexing errors, and other common programming errors. Moreover, Rust's built-in unit testing is used to ensure each function and module produces the correct outputs given the known inputs. To accelerate our code, we used Arrayfire for Rust[46], a parallel computing library for CUDA, OpenCL, and OpenMP devices. This enables the software framework to run on Nvidia GPUs/CPUs, AMD GPUs/CPUs, Intel GPUs/CPUs, and Xilinx FPGAs.

## Supplementary information

We wrote a supplementary material article[25] describing the specific implementation details of the RayBNN. We also included an animation showing the evolution of the RayBNN while training on the Alcala dataset.

## Reporting summary

Further information on research design is available in the Nature Portfolio Reporting Summary linked to this article.

## Data availability

There are no datasets generated during and/or analyzed during the current study. All datasets used in the study are publicly available at [32,35]. Source data are available at https://www.sensor-net.net/a-3d-ray-traced-biological-neural-network-learning-model/.

## Code availability

The code is available at https://www.sensor-net.net/a-3d-ray-traced-biological-neural-network-learning-model/.

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

## Acknowledgements

This project is supported by the Natural Sciences and Engineering Research Council of Canada (NSERC) Discovery Grants No. RGPIN-2018-03778 (X.D.), and RGPIN-2020-05938 (T.L.), NSERC Alliance Grant No. ALLRP 571684-21 (X.D. and T.L.), and Defense Threat Reduction Agency (DTRA) Thrust Area 7, Topic G18 Grant No. GRANT12500317 (T.L.). This research was enabled in part by support provided by SFU/Cedar supercomputer (cedar.computecanada.ca) and the Digital Research Alliance of Canada (https://alliancecan.ca).

## Author contributions

B.Y. and T.L. conceptualized the work and revised the idea for intellectual content. B.Y. wrote the manuscript. T.L. and X.D. performed substantial editorial work. B.Y. and T.L. implemented the idea. B.Y. implemented software framework. T.L. and X.D. coordinated and supervised the work.

## Competing interests

The authors declare no competing interests.
