## [Peer Review File · Nature Communications]

REVIEWER COMMENTS

Reviewer #2 (Remarks to the Author):

This manuscript introduces a novel neural network approach inspired by biological neural networks to address the limitations of current transfer learning algorithms. The primary motivation is to reduce the computational resources and time required for training large neural networks on big datasets. The authors propose a unique neural network architecture that is highly transferable across various network architectures and adaptable to multiple datasets. This innovative approach leverages raytracing to connect neurons in a three-dimensional space, allowing the network to dynamically grow and reshape itself to accommodate different problem domains and input sizes. The manuscript highlights the effectiveness of their transfer learning algorithm on the Alcalá dataset, showcasing its ability to train quickly and efficiently in changing environments and with varying input sizes. Additionally, the authors suggest that this novel network design could have implications for real biological neural networks, potentially leading to reduced power consumption in future implementations. In summary, the manuscript presents a cutting-edge neural network architecture inspired by biological neural networks, demonstrating its potential to significantly improve transfer learning capabilities and its suitability for various datasets and changing environments.

This manuscript is truly outstanding and represents a remarkable contribution to the field of neural networks and transfer learning. The innovative approach inspired by biological neural networks is both creative and promising. By addressing the limitations of current transfer learning algorithms and allowing neural networks to adapt dynamically to different problem domains and input sizes, this manuscript opens up exciting possibilities for more efficient and effective machine learning applications. The authors' use of raytracing to connect neurons in a three-dimensional space is a brilliant concept that offers a refreshing perspective on neural network architecture design. Their success in demonstrating the algorithm's efficiency on the Alcalá dataset underscores the practical value of their approach. Furthermore, the manuscript's forward-looking vision of potentially implementing this novel network design in real biological neural networks to reduce power consumption is nothing short of visionary. It highlights the potential real-world impact of their research. Overall, this manuscript is a testament to the power of innovative thinking and the potential for groundbreaking advancements in the field of machine learning. It deserves high praise for its creativity, practicality, and forward-looking perspective.

I think about the critical role that space location plays in biological neural networks. It would be intriguing to explore how your innovative approach, which allows neural networks to dynamically grow and reshape themselves, might be influenced by the spatial organization of neurons in biological systems. Have you considered the potential impact of spatial arrangements on the performance and adaptability of your neural network design? This could further enhance the relevance and applicability of your research. Streamlining the language and exploring the impact of spatial arrangements on your model's adaptability could enhance the clarity and relevance of your research.

In section 3.9, please explain the essential differences between your method and Dropout, and conduct comparative experiments on them. In addition, there are too many magic numbers in the manuscript, please explain why you choose that number?

Reviewer #4 (Remarks to the Author):

Transfer learning is a widely used framework in many neural network applications. It facilitates transfer of knowledge, acquired initially from training on large datasets, to related downstream tasks for which data are typically much more limited. (The initial training phase is sometimes called “pretraining” and the latter transfer phase “fine tuning”). A common constraint among current transfer learning methods is their dependency on the network architecture that was used in the pre-training phase. To address this constraint, the authors propose RayBNN, a machine learning approach inspired by biological neural networks from the nervous system. RayBNN emulates the interactions between neurons and glial cells, allowing flexible neural network architecture during transfer learning.

The concept of emulating a biological neural network to enhance the transfer learning framework is intriguing. However, we are concerned that the work as it now stands is not sufficient to support the significance or utility of the proposed method.

1. First and foremost, the methods have only been validated on a single small-scale dataset (Alcala Tutorial 2017 dataset), raising important questions about generalizability.
2. Second, the improvements in computational time and performance reported, albeit positive, seem marginal and do not justify the transition to this new approach without further evidence of substantial improvement in additional, larger-scale, datasets.
3. Third, the authors should clarify which functions are borrowed from the biological neural network (by providing references) and justify why they are related to enhancing transfer learning. Such clarification should be supported by clear examples and illustrations, e.g. in Figure 1.

In conclusion, the paper as it stands does not convincingly establish a novel contribution to the field of transfer learning.

Author Responses

Re: NCOMMS-23-36672-T

Article Title: "RayBNN: A 3-D Biological Neural Network Transfer Learning Model"

Response to the Reviewer #1:

Comment #1-1:

This manuscript introduces a novel neural network approach inspired by biological neural networks to address the limitations of current transfer learning algorithms. The primary motivation is to reduce the computational resources and time required for training large neural networks on big datasets. The authors propose a unique neural network architecture that is highly transferable across various network architectures and adaptable to multiple datasets. This innovative approach leverages raytracing to connect neurons in a three-dimensional space, allowing the network to dynamically grow and reshape itself to accommodate different problem domains and input sizes. The manuscript highlights the effectiveness of their transfer learning algorithm on the Alcalá dataset, showcasing its ability to train quickly and efficiently in changing environments and with varying input sizes. Additionally, the authors suggest that this novel network design could have implications for real biological neural networks, potentially leading to reduced power consumption in future implementations. In summary, the manuscript presents a cutting-edge neural network architecture inspired by biological neural networks, demonstrating its potential to significantly improve transfer learning capabilities and its suitability for various datasets and changing environments.

This manuscript is truly outstanding and represents a remarkable contribution to the field of neural networks and transfer learning. The innovative approach inspired by biological neural networks is both creative and promising. By addressing the limitations of current transfer learning algorithms and allowing neural networks to adapt dynamically to different problem domains and input sizes, this manuscript opens up exciting possibilities for more efficient and effective machine learning applications. The authors' use of raytracing to connect neurons in a three-dimensional space is a brilliant concept that offers a refreshing perspective on neural network architecture design. Their success in demonstrating the algorithm's efficiency on the Alcalá dataset underscores the practical value of their approach. Furthermore, the manuscript's forward-looking vision of potentially implementing this novel network design in real biological neural networks to reduce power consumption is nothing short of visionary. It highlights the potential real-world impact of their research. Overall, this manuscript is a testament to the power of innovative thinking and the potential for groundbreaking advancements in the field of machine learning. It deserves high praise for its creativity, practicality, and forward-looking perspective.

Author response #1-1:

We thank you for the encouraging comments. We hope our RayBNN will contribute to the interdisciplinary research among AI, neurological, and photonics societies. We hope that our continuing research on this subject will help our understanding of nature and life.

Comment #1-2:

I think about the critical role that space location plays in biological neural networks. It would be intriguing to explore how your innovative approach, which allows neural networks to dynamically grow and reshape themselves, might be influenced by the spatial organization of neurons in biological systems. Have you considered the potential impact of spatial arrangements on the performance and

adaptability of your neural network design? This could further enhance the relevance and applicability of your research. Streamlining the language and exploring the impact of spatial arrangements on your model's adaptability could enhance the clarity and relevance of your research.

Author response #1-2:

We thank you for the inspiring comment. We thought through the idea of arranging the neurons and glial cells in a patterned fashion. This is truly an interesting idea and shares a lot of resemblance to the group theory and solid state physics where a lot of research on how atoms/molecules are arranged in a patterned topology has been done. We now believe some neuron topology may be particularly suitable for a subset of applications. This will be an exciting future research.

This comment also inspires our thinking on the possible impacts of network shape on the overall performance of the RayBNN to specific applications. We believe it will be another exciting research in the future to automatically evolve our network shape through training.

Action #1-2:

We have added a few paragraphs on Page 15 of the main text under Section 2 to address these points.

2 Discussion

In this article, we randomly positioned neurons in a 3D sphere. As shown in Fig. 2e, the probability density function of the neuron lengths is a continuous Gaussian curve. This gives a lot of flexibility for creating many different neural connections and neural network structures. Alternatively, neurons may be arranged in a patterned fashion. For example, when neurons are arranged on a set of concentric sphere surfaces and only allow neural connections between neighboring surfaces, then the RayBNN topology becomes equivalent to a conventionally layered neural network. Overall, there are many possible periodic or chaotic arrangements for neurons and glial cells. It is possible that certain arrangements, along with certain connection rules, will lead to out-performance over the state of the art in a set of applications. It is also feasible to optimize the position of neurons and glial cells through training. Therefore, implementing them and exploring their characteristics will be exciting research in the future. In particular, one may study it with the knowledge transformed from group theory and solid state physics where various spatial topologies of atoms and molecules in solid have been extensively investigated.

The network's physical shape could be another exploration factor as it heavily influences the neural connection length and in turn, also influences the propagation delay of information. For example, a neural network's overall shape could be a 3D cube, a torus, an ellipsoid, or a tetrahedron, etc., each with its own advantages and disadvantages. In the case of a 3D cube or tetrahedron, the signal paths from the surface of the shape to the center are very different compared to a 3D sphere. As a result, some signals may reach the center of a cube faster, while others might take a longer time since they traveled a longer path. For applications needing synchronization, 3D cube and tetrahedron networks might not be optimal. For other applications requiring joining or merging neural networks, 3D cube, and tetrahedron networks can be easily joined together to significantly scale up a neural network for large datasets. In the case of an ellipsoid, the shape changes the number of hidden neurons that each input neuron can access as it now becomes surface location dependent. Such bias toward a certain subset of input data features may be advantageous to train datasets whose data features are not equally important. It may be possible to evolve the network shape toward an ergodic optimal according to the nature of the dataset, just like how the human brain evolves from a sphere [40].

Comment #1-3:

In section 3.9, please explain the essential differences between your method and Dropout, and conduct comparative experiments on them.

Author response #1-3:

We thank you for the comment. We made changes to Section 3.9 explaining the differences between dropout and deleting redundant neurons. We also conducted comparative experiments on the new data set.

Action #1-3:

1. We have added one paragraph in Section 3.9 for the explanation on Page 29 of the main text.

The difference between deleting redundant neurons and dropout is that dropout randomly deletes neurons and neural connections, of which changes the outputs of the neural network/layer. On the other hand, deleting redundant neurons does not change the outputs of the neural network because the redundant neurons are not outputting any information into other neurons.

2. We also compared the dropout vs. deleting redundant neurons with the new EEG datasets and plotted them as Fig. 5b on Page 12 of the main text.

(b)

Fig. 5: (a) RayBNN transfer learning for EEG dataset, and (b) Comparison of RayBNN and MLP in the EEG Dataset [34]. (c) EEG dataset and OpenBMI Toolbox for Three BCI Paradigms: An Investigation into BCI Illiteracy [34]. 54-fold testing. Motor imagery classification.

3. We added a paragraph to describe the comparison on Page 14 of the main text.

Fig. 5b shows a comparison between the Xdawn-Deep4Net-RayBNN and its Xdawn-Deep4Net-MLP counterpart for one of the testing folds in the EEG dataset. The MLP has a dropout rate of 50% and the RayBNN has a sparsity of approximately 50%. As the number of trainable parameters increases, the ROC AUC also increases. However, the ROC AUC eventually reaches a limit, even though the number of trainable parameters keeps increasing. As shown in the figure, RayBNN performs much better than MLP due to having neural connection pruning and deleting redundant neurons. Fig. 5c shows the performances of the algorithms on an individual subject basis. The Xdawn algorithm performs better for some subjects than the Deep4Net. Conversely, Deep4Net performs better for some subjects than the Xdawn algorithm. Due to the fact RayBNN uses both Xdawn and Deep4Net, it has the advantages of both and produces the highest accuracy for most of the test index. For the training time of the various algorithms, the CSP-LDA algorithm has a training time of 15.73 ± 0.91 seconds and CSP-LR has 15.51 ± 0.97 seconds. Moreover, Xdawn-MDM and Xdawn-LR have 19.21 ± 1.2 seconds and 19.05 ± 1.6 seconds respectively. On the other hand, Deep4Net has a training time of $7,271 \pm 231$ seconds, which is drastically higher. Subsequently, Xdawn-Deep4Net-MLP, Deep4Net-RayBNN, Xdawn-Deep4Net-RayBNN have $7,324 \pm 235$ s and $7,306 \pm 233$ s and 7326 ± 235 s respectively due to the incorporation of Deep4Net.

Table 2 shows the statistical testing of each EEG algorithm in comparison to Xdawn-Deep4Net-RayBNN. The accuracy is calculated for each individual algorithm and fold. To compare, we select two algorithms and compute the difference in accuracy for each fold. We applied the paired t-test to the differences to get the p-values. The null hypothesis assumes the difference between the algorithms has a mean equal to zero. As all p-values are equal or less than 1.7968×10^{-3} , we reject the null hypothesis and assert the Xdawn-Deep4Net-RayBNN is statistically better than all of the other algorithms.

Table 2: Statistical Testing of the EEG algorithms.

Comparison	p-value
Xdawn-Deep4Net-RayBNN vs Xdawn-Deep4Net-MLP	1.2725×10^{-4}
Xdawn-Deep4Net-RayBNN vs Deep4Net	1.7968×10^{-3}
Xdawn-Deep4Net-RayBNN vs Xdawn-LR	2.2429×10^{-6}
Xdawn-Deep4Net-RayBNN vs Xdawn-MDM	1.0220×10^{-16}
Xdawn-Deep4Net-RayBNN vs CSP-LR	2.3756×10^{-24}
Xdawn-Deep4Net-RayBNN vs CSP-LDA	2.2413×10^{-24}

Note: Paired t-test with right tail p-values of the accuracy. 54-fold testing.

Comment #1-4:

In addition, there are too many magic numbers in the manuscript, please explain why choose that number?

Author response #1-4:

We thank you for the comment. Using the Alcalá dataset as an example, we now standardize the justifications to determine these magic numbers.

Action #1-4:

We have revised the text in Section 1.3 from the bottom of Page 9 on how to determine the network parameters and build RayBNN from it.

To simulate this, we started with 6 APs as our initial training dataset and built our initial RayBNN upon it. The initial RayBNN has 6 input neurons, 2 output neurons. Although the number of hidden neurons can be determined through a standard hyperparameter tuning process, we here empirically set it to 40. Correspondingly, we assign an equal number of glial cells to mimic the real biological neural network although it can also be tuned if necessary. With the prescribed algorithm in Section 3.2.1, the network sphere is set to $r_s = 42r_n$ to keep the collision rate below 1%. Consequently, through the RT-3, 1,800 connections are created with a total of 5,300 trainable parameters. After training, we increased the dimension of the new training dataset empirically to 8 APs and transferred the trained model to the new dataset. As every AP provides 1 input feature, the number of neural network inputs of the new dataset increases along with the model complexity. Therefore, we increased the network to 8 input neurons. Following the same procedure as the previous iteration, we also increase the network to 50 hidden neurons, and 50 glial cells, while adjusting the network sphere to $r_s = 45r_n$ accordingly. Meanwhile, 5,700 new connections are also created before training, leading to the total number of parameters to 11,000. As shown in the red circles with a solid red line in Fig. 4a, this process continued until the network reached the maximum input feature size of 162.

Response to the Reviewer #2:

Comment #2-1:

Transfer learning is a widely used framework in many neural network applications. It facilitates transfer of knowledge, acquired initially from training on large datasets, to related downstream tasks for which data are typically much more limited. (The initial training phase is sometimes called “pretraining”; and the latter transfer phase “fine tuning”). A common constraint among current transfer learning methods is their dependency on the network architecture that was used in the pre-training phase. To address this constraint, the authors propose RayBNN, a machine learning approach inspired by biological neural networks from the nervous system. RayBNN emulates the interactions between neurons and glial cells, allowing flexible neural network architecture during transfer learning.

The concept of emulating a biological neural network to enhance the transfer learning framework is intriguing. However, we are concerned that the work as it now stands is not sufficient to support the significance or utility of the proposed method.

1. First and foremost, the methods have only been validated on a single small-scale dataset (Alcala Tutorial 2017 dataset), raising important questions about generalizability.

Author response #2-1:

We thank you for the critical comment. In the current revision, we added a large dataset: a 210-GB EEG dataset from [34] for more comprehensive assessment of the RayBNN.

Also following the discussions in Response #1-2, the conventional artificial neural network is a subset of RayBNN under specific neuron arrangement and connection rules: neurons are located at concentric sphere surfaces and we only allow connections of neurons between neighboring surfaces. Therefore, it is evident that RayBNN is more generalizable than conventional artificial neural networks or at least as generalizable as it.

Action #2-1:

We have added the discussions on the new EEG dataset in Section 1.4 starting from page 11.

1.4 EEG motor imagery dataset

In EEG datasets, the objective is to retrieve information from the subject’s brain using multiple electrodes placed on the subject’s head/brain. However, every human has a unique set of EEG signals that is completely different from every other person. This is due to having distinct brain structures and electrode placements. As a consequence, most algorithms are unable to perfectly generalize across different subjects, especially if they have not seen the subject’s specific waveforms before.

Table 1 shows the algorithms’ performances on a 210-GB EEG dataset [34]. In this dataset, there are 54 different subjects and each subject has two experimental sessions for classifying and detecting motor-imagery (MI) tasks, event-related potential (ERP), and steady-state visually evoked potential (SSVEP) tasks. 54-fold subject-independent testing is used to evaluate the models in Table 1. For each fold, 1 subject is selected for the testing dataset, while the other 53 subjects are selected for the training dataset to remove any overlap between the training dataset and the testing dataset. Moreover, there are no duplicate samples between the testing datasets in each fold. That way, the algorithms are evaluated on their ability to generalize across subjects. Accuracy, precision, recall, F_1 score, and area under curve receiver operating characteristic (AUC ROC) are recorded for the various algorithms.

Common spatial pattern (CSP) [35, 36] is widely used for extracting EEG features by decomposing the multivariate EEG signal into component eigenvalues and eigenvectors. After extracting the features, they are fed into linear discriminant analysis (LDA) or logistic regression (LR) for classification. As shown in Table 1, CSP-LDA is not very good at generalizing across different subjects for this specific dataset and has a very low mean accuracy of 62.4%. CSP-LR has a slightly better accuracy of 62.5%. On the other hand, researchers have used the Xdawn algorithm [37] from the pyRiemann python package [38] to extract features from EEG signals. Xdawn projects the high-dimensional Riemann manifold source space to the tangent space, which allows each class to be discerned more easily than the source space. Subsequently, the minimum distance to mean (MDM) algorithm is used to produce the final classification result. Each class has a centroid, and the data samples closest to a specific centroid will be assigned to that specific class. The combination of Xdawn and MDM (Xdawn-MDM) performs significantly better than CSP algorithms, as its accuracy of 71.2% is much higher. Furthermore, using Xdawn-LR increases the accuracy to 82.7%.

Deep4Net [39] was developed as the state-of-the-art CNN model for classifying EEG signals, of which is made out of 5 blocks. Each block has a 2D convolutional layer, batch normalization layer, max pooling layer, and dropout layer. Moreover, the model does not have any fully connected layers but uses a logsoftmax function as its final layer. Deep4Net’s 83.6% accuracy is higher than Xdawn’s accuracy because the convolutional layers can denoise and extract more features than the Xdawn algorithm. To outperform the state-of-the-art, we incorporate RayBNN together with Deep4Net as shown in Fig. 5a. Since Deep4Net’s final layer aggregates data and loses a lot of information, we extract outputs from Deep4Net’s second last layer and feed it into RayBNN’s input neurons. For RayBNN’s architecture, there are 1,400 input neurons, 1,000 hidden neurons, and 600,000 neural connections. Subsequently, RayBNN produces the final classification result for the EEG dataset. For the Deep4Net-RayBNN combination, it has an accuracy of 84.6% which is higher than standalone Deep4Net and Xdawn-Deep4Net-MLP. As there is no optimal feature extraction algorithm for all subjects, we decided to create an ensemble of Xdawn-Deep4Net-RayBNN as shown in Fig. 5a. This is done by first training the Deep4Net-RayBNN combination and transferring the network to the Xdawn-Deep4Net-RayBNN ensemble. The transfer learning flexibility of RayBNN allows it to dynamically accept the 1,400-element output from Deep4Net and the 990-element output from Xdawn to predict the final EEG classification result. For this specific case, the RayBNN has 2,390 input neurons, 1,000 hidden neurons, and 600,000 neural connections. Overall, the Xdawn-Deep4Net-RayBNN ensemble has the highest accuracy of 85.6% with precision, recall, F_1 score, and AUC ROC being higher than the rest of the algorithms.

Fig. 5b shows a comparison between the Xdawn-Deep4Net-RayBNN and its Xdawn-Deep4Net-MLP counterpart for one of the testing folds in the EEG dataset. The MLP has a dropout rate of 50% and the RayBNN has a sparsity of approximately 50%. As the number of trainable parameters increases, the ROC AUC also increases. However, the ROC AUC eventually reaches a limit, even though the number of trainable parameters keeps increasing. As shown in the figure, RayBNN performs much better than MLP due to having neural connection pruning and deleting redundant neurons. Fig. 5c shows the performances of the algorithms on an individual subject basis. The Xdawn algorithm performs better for some subjects than the Deep4Net. Conversely, Deep4Net performs better for some subjects than the Xdawn algorithm. Due to the fact RayBNN uses both Xdawn and Deep4Net, it has the advantages of both and produces the highest accuracy for most of the test index. For the training time of the various algorithms, the CSP-LDA algorithm has a training time of 15.73 ± 0.91 seconds and CSP-LR has 15.51 ± 0.97 seconds. Moreover, Xdawn-MDM and Xdawn-LR have 19.21 ± 1.2 seconds and 19.05 ± 1.6 seconds respectively. On the other hand, Deep4Net has a training time of $7,271 \pm 231$ seconds, which is drastically higher. Subsequently, Xdawn-Deep4Net-MLP, Deep4Net-RayBNN, Xdawn-Deep4Net-RayBNN have $7,324 \pm 235$ s and $7,306 \pm 233$ s and 7326 ± 235 s respectively due to the incorporation of Deep4Net.

Table 2 shows the statistical testing of each EEG algorithm in comparison to Xdawn-Deep4Net-RayBNN. The accuracy is calculated for each individual algorithm and fold. To compare, we select two algorithms and compute the difference in accuracy for each fold. We applied the paired t-test to the differences to get the p-values. The null hypothesis assumes the difference between the algorithms has a mean equal to zero. As all p-values are equal or less than 1.7968×10^{-3} , we reject the null hypothesis and assert the Xdawn-Deep4Net-RayBNN is statistically better than all of the other algorithms.

Fig. 5: (a) RayBNN transfer learning for EEG dataset, and (b) Comparison of RayBNN and MLP in the EEG Dataset [34]. (c) EEG dataset and OpenBMI Toolbox for Three BCI Paradigms: An Investigation into BCI Illiteracy [34]. 54-fold testing. Motor imagery classification.

Table 1: Performances of the Algorithms in the EEG Motor Imagery Dataset [34].

Model	Accuracy	Precision	Recall	F_1 score	ROC AUC
CSP-LDA [35, 36]	0.624 ± 0.092	0.638 ± 0.097	0.624 ± 0.092	0.609 ± 0.103	0.646 ± 0.121
CSP-LR [35, 36]	0.625 ± 0.092	0.639 ± 0.097	0.625 ± 0.092	0.610 ± 0.103	0.646 ± 0.121
Xdawn-MDM [37, 38]	0.712 ± 0.112	0.732 ± 0.106	0.712 ± 0.112	0.701 ± 0.123	0.770 ± 0.129
Xdawn-LR [37, 38]	0.827 ± 0.087	0.835 ± 0.083	0.827 ± 0.087	0.826 ± 0.089	0.891 ± 0.083
Deep4Net [39]	0.836 ± 0.108	0.851 ± 0.094	0.836 ± 0.108	0.831 ± 0.121	0.914 ± 0.086
Xdawn-Deep4Net-MLP	0.836 ± 0.085	0.844 ± 0.081	0.836 ± 0.085	0.834 ± 0.086	0.920 ± 0.071
Deep4Net-RayBNN	0.846 ± 0.104	0.849 ± 0.103	0.846 ± 0.104	0.845 ± 0.104	0.906 ± 0.094
Xdawn-Deep4Net-RayBNN	0.856 ± 0.085	0.861 ± 0.082	0.856 ± 0.085	0.856 ± 0.086	0.926 ± 0.068

Note: 54-fold testing. Subject-Independent. Confidence interval of 1σ .

Table 2: Statistical Testing of the EEG algorithms.

Comparison	p-value
Xdawn-Deep4Net-RayBNN vs Xdawn-Deep4Net-MLP	1.2725×10^{-4}
Xdawn-Deep4Net-RayBNN vs Deep4Net	1.7968×10^{-3}
Xdawn-Deep4Net-RayBNN vs Xdawn-LR	2.2429×10^{-6}
Xdawn-Deep4Net-RayBNN vs Xdawn-MDM	1.0220×10^{-16}
Xdawn-Deep4Net-RayBNN vs CSP-LR	2.3756×10^{-24}
Xdawn-Deep4Net-RayBNN vs CSP-LDA	2.2413×10^{-24}

Note: Paired t-test with right tail p-values of the accuracy. 54-fold testing.

Comment #2-2:

2. Second, the improvements in computational time and performance reported, albeit positive, seem marginal and do not justify the transition to this new approach without further evidence of substantial improvement in additional, larger-scale, datasets.

Author response #2-2:

We thank you for the comment. At the current state, significantly outperforming the conventional neural network is not our primary goal. We agree that there is room for improvements for RayBNN as discussed in the response to Comment #1-2, but the RayBNN already shows much faster training time in the Alcalá dataset and topped the performance to the state-of-the-art for both Alcalá and EEG datasets. In the EEG dataset, we further showed that transfer learning is fully automated for switching between different inputs and architectures, and there is an advantage to using RayBNN as an ensemble method for quickly adapting to new output tensor sizes from other machine learning algorithms.

Following discussions in Response #1-2, we will further optimize the RayBNN for performance enhancement. The fact that conventional artificial neural network is a subset of RayBNN and existing deep neural network can be seamlessly incorporated into RayBNN in the example of EEG dataset supports the perspective that RayBNN may significantly outperform existing ML models both on training time and performance metrics such as accuracy, precision, recall, F_1 score, etc.

Most importantly, we are not stopped at just developing a better new machine learning model from an algorithmic perspective. The resemblance of RayBNN to real biological neural networks provides

us with a unique platform to study human and animal intelligence and their behavior. Since RayBNN's structure is physically constrained to 3D, the RayBNN's network models can be physically implemented in a real 3D neuromorphic computer or a real biological neural network, which promises lower power consumption and similar behavior of human intelligence. Such investigations benefit both scientists and engineers who enjoy understanding life and building powerful tools from it.

Action #2-2:

- a) See Action #1-2 for future research on RayBNN improvements.
- b) We revised the discussions on Page 16

Overall, we created an entirely new three-dimensional RayBNN transfer learning model that is a close similarity to real-life BNN. In the world of machine learning, the traditional artificial neural network (ANN) is usually planar with well-structured neural network layers. Our RayBNN, like real-life BNN and unlike ANN, assigns 3-D positions to neurons and glial cell in a neural network sphere. The neurons are interconnected stochastically without well-defined layers, allowing information flow and learning transfers more efficient. Although still in its infant stage, our RayBNN has already outperformed conventional models in indoor localization, on both speed and accuracy. It also tops the state of the art in large EEG dataset analysis and predictions and demonstrates its capacity for seamless integration with conventional deep neural networks, which brings additional power to it. Note that up to date, the human brain still out-beats AI in many aspects, such as using symbolic logic to derive mathematical proofs, handling numerous incompatible data structures, and achieving multiple different objectives at the same time. We expect that with the continuing development, RayBNN will out-beat other AI models in these areas due to its inherent similarity to BNN.

As a human brain consumes much less power than current AI models, we intend to use real-life neurons or in particular, optogenetically modified neurons, to implement our RayBNN so that the network can be trained and the input/output be read in/out optically, which may lead to a better AI hardware but with much lower power.

Our research on RayBNN will not stop at developing new better machine learning algorithms and hardware. The resemblance between the RayBNN and real-life biological neural networks makes RayBNN a unique platform for the studies of human and animal intelligence and behavior. With further development, the RayBNN neuromorphic device may be miniaturized so that it can be trained and implemented to patients for neural disease treatments.

Comment #3:

3. Third, the authors should clarify which functions are borrowed from the biological neural network (by providing references) and justify why they are related to enhancing transfer learning. Such clarification should be supported by clear examples and illustrations, e.g. in Figure 1.

In conclusion, the paper as it stands does not convincingly establish a novel contribution to the field of transfer learning.

Author response #2-3:

We thank you for the comment. We added a paragraph in the discussion that addresses the comment above.

Action #2-3:

We have added a paragraph on Page 5 right above Section 1.2.

RayBNN is very similar to real-life biological neural networks due to having 3D physical cell locations, line-of-sight neural connectivity, signal propagation delays, glial cells, cell growth, cell death, neural network merges, and neural network bifurcations. Firstly, both the RayBNN and the real-life BNN are physically constrained by the radius of the entire neural network, cell radii, and cell density. For a neural network radius, there is a finite amount of cells within the volume because the cells can not be closer than 2 cell radii. Due to those physical constraints, both RayBNN and real-life BNN have line-of-sight neural connections that can be blocked by glial cells or other neurons. Subsequently, RayBNN has a signal propagation delay that is similar to a real-life BNN because it takes time for information to travel from one neuron to another. Real-life BNN has glial cells to inhibit or to electrically isolate neurons from each other to prevent infinite signal loops or neuron overfiring. With the same idea, we implemented glial cells in our RayBNN to reduce neural connections and to prevent overfitting of the network. Similar to real-life BNN, our RayBNN can dynamically grow or shrink by adding new neurons or deleting neurons. Moreover, our RayBNN can join or merge multiple neural networks different along multiple axes. This has a higher degree of connectivity between blocks than traditional artificial neural networks and results in better integrations.

REVIEWERS' COMMENTS

Reviewer #2 (Remarks to the Author):

The revised paper addresses my concerns, and I am satisfied with the author's responses. The author also provided detailed information. I believe this version of the paper is ready for publication.

Reviewer #4 (Remarks to the Author):

I appreciate the authors revisions and especially their addition of a large new EEG dataset which helps demonstrate the generalizability of their approach. No further comments.